# Res-Tuning: A Flexible and Efficient Tuning Paradigm via Unbinding Tuner from Backbone

**Zeyinzi Jiang**[1]     **Chaojie Mao**[1]     **Ziyuan Huang**[2]     **Ao Ma**[1]
**Yiliang Lv**[1]     **Yujun Shen**[3]     **Deli Zhao**[1]     **Jingren Zhou**[1]

[1]Alibaba Group     [2]National University of Singapore     [3]Ant Group
{zeyinzi.jzyz, chaojie.mcj, dave.ma, yiliang.lyl, jingren.zhou}@alibaba-inc.com
{ziyuan.huang}@u.nus.edu {shenyujun0302, zhaodeli}@gmail.com

## Abstract

Parameter-efficient tuning has become a trend in transferring large-scale foundation models to downstream applications. Existing methods typically *embed* some light-weight tuners into the backbone, where both the design and the learning of the tuners are highly dependent on the base model. This work offers a new tuning paradigm, dubbed `Res-Tuning`, which intentionally *unbinds* tuners from the backbone. With both theoretical and empirical evidence, we show that popular tuning approaches have their equivalent counterparts under our unbinding formulation, and hence can be integrated into our framework effortlessly. Thanks to the structural disentanglement, we manage to free the design of tuners from the network architecture, facilitating flexible combination of various tuning strategies. We further propose a memory-efficient variant of `Res-Tuning`, where the bypass (*i.e.*, formed by a sequence of tuners) is effectively detached from the main branch, such that the gradients are back-propagated only to the tuners but not to the backbone. Such a detachment also allows one-time backbone forward for multi-task inference. Extensive experiments on both discriminative and generative tasks demonstrate the superiority of our method over existing alternatives from the perspectives of efficacy and efficiency. Project page: https://res-tuning.github.io/.

## 1   Introduction

Recently, foundation models have demonstrated strong generalization capability across numerous visual [21, 2], language [47, 60] and multi-modal tasks [40, 1]. Pre-trained on a large corpus of data, a foundation model offers a good initialization for downstream adaptation. Unfortunately, the increasing model scale makes it expensive and almost infeasible to fully fine-tune such a model for every task. Hence, parameter-efficient transfer learning (PETL) [37, 41, 27] is often resorted to as an efficient approach for downstream adaptation without incurring an unaffordable computation burden.

Popular existing approaches for parameter-efficient tuning introduce additional tunable structures (which we term as *tuners*) to the pre-trained base model [37, 41, 27]. Compared to the fully-fine-tuned counterparts, the light-weight tuners significantly reduce the training cost while maintaining a competitive performance [32, 28]. However, the current designs of tuners are deeply coupled with their base structures, as shown in Fig. 1a, thus restricting the design flexibility and impeding the extension to new approaches. For example, prefix tuning [41] is embedded into the self-attention operation, and prompts [28, 85] could only be introduced at the beginning or between layers, *etc*.

In this work, we introduce `Res-Tuning`, a new tuning paradigm for flexible and efficient transfer learning. As in Fig. 1b, our `Res-Tuning` framework unbinds the tuners from the base model, such that it is possible to decouple both the design and the learning of the tuners from the base structure. Since

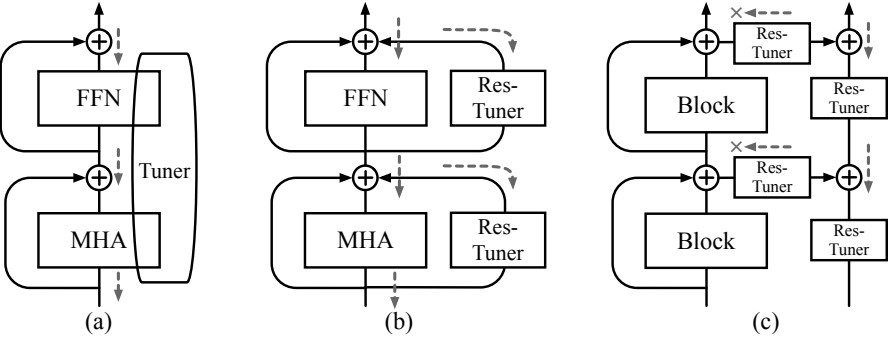

Figure 1: **Concept comparison** between existing methods and our `Res-Tuning` framework. (a) Existing PETL methods are deeply embedded into original structures. (b) Our `Res-Tuning` framework can decouple PETL methods from the backbone. (c) Backpropagation only through a bypass consisting of `Res-Tuner` can be achieved by detaching the connections.

the design of the tuners is no longer dependent on the base structure in our `Res-Tuning` framework, we explore various possibilities. With both theoretical and empirical evidence, we first show that our framework can seamlessly encompass popular tuning approaches such as prefix-tuning [41], prompt-tuning [28], and adapters [25]. Further, it is demonstrated that the structural disentanglement also allows for flexible combination of various tuners, leading to the discovery of stronger tuning strategies. On top of that, we show that such an unbinding formulation also allows for the detachment of the tuners from the backbone as in Fig. 1c, which further improves the memory efficiency. In this memory-efficient version, *i.e.,* `Res-Tuning-Bypass`, not only is the training cost reduced because the gradient computation through the massive parameters is avoided in the base model, but it also reduces the number of forward passes of the backbone model to only once during multi-task inference.

We evaluate `Res-Tuning` framework on both discriminative and generative tasks. On discriminative tasks, we show that our unbinding formulation leads to a tuning strategy that achieves state-of-the-art performance on VTAB-1K with similar learnable parameters. Our `Res-Tuning-Bypass` framework also performs favorably against the fully-fine-tuned variant, while reducing the training memory consumption by 49.7% and the multi-task inference time by 80.9%. In addition, it also obtains better performance in few-shot learning and domain generalization scenarios. On generative tasks, apart from the strong performance achieved by `Res-Tuning` framework in terms of both FID scores and qualitative results, we further show that our `Res-Tuning-Bypass` can reduce the memory consumption by 70.7%, training time by 58.6%, and multi-task inference time by 83.6% when maintaining a competitive FID score and generation quality compared to the fully-fine-tuned variant.

## 2 Unbinding parameter-efficient tuning

In order to reduce the entanglement between the base model and the tuners as well as to increase the flexibility of the tuning strategies, we set out to unbind the tuners from the pre-trained structures. In this section, we provide our unbinding formulation to existing parameter-efficient transfer learning strategies. We start by revisiting the basic building blocks of the foundation models, before diving into unbinding existing tuners from the backbone. In the last part of this section, we further provide empirical proof that our unbinding formulation could seamlessly encompass existing PETL methods like prompt tuning [28], prefix tuning [41], and adapters [25].

### 2.1 Basic building blocks of foundation models

Existing foundation models in both natural language processing, vision, and vision-language applications mostly adopt Transformers [70] as the backbone. The major building blocks of the Transformers that one usually adapts for the downstream tasks are the multi-head attention (MHA) and the feed-forward network (FFN). Formally, the standard MHA and FFN could be expressed as:

$$\text{Attn}(\boldsymbol{Q}, \boldsymbol{K}, \boldsymbol{V}) = \text{softmax}\left(\frac{\boldsymbol{Q}\boldsymbol{K}^{\text{T}}}{\sqrt{d}}\right)\boldsymbol{V} \ ,$$
$$\text{FFN}(\boldsymbol{x}) = \phi(\boldsymbol{x}\boldsymbol{W}_1 + \boldsymbol{b}_1)\boldsymbol{W}_2 + \boldsymbol{b}_2 \tag{1}$$

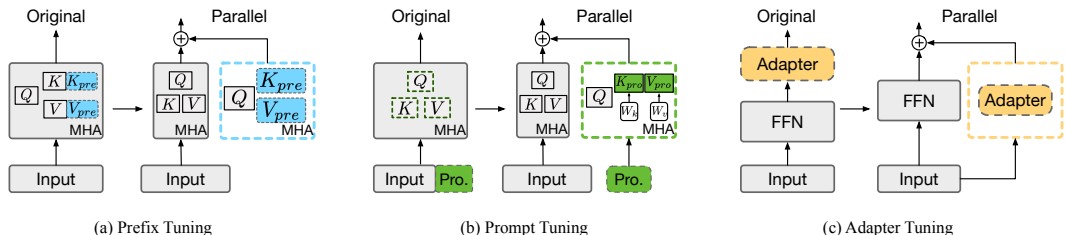

Figure 2: **The original and the unbinding form** of (a) prefix tuning [41], (b) prompt tuning [28], and (c) adapter tuning [25] for parameter-efficient transfer learning.

where $\boldsymbol{Q}$, $\boldsymbol{K}$ and $\boldsymbol{V}$ denote the query, key and value, respectively. $\boldsymbol{W}_1$ and $\boldsymbol{W}_2$ are the projection weights, $\boldsymbol{b}_1$ and $\boldsymbol{b}_2$ are the bias terms, and $\phi$ is the non-linear activation between fully-connected layers. Usually, given input tokens $\boldsymbol{x}$, the query, key, and value are obtained through a linear projection $\boldsymbol{Q} = \boldsymbol{x}\boldsymbol{W}_q$, $\boldsymbol{K} = \boldsymbol{x}\boldsymbol{W}_k$, and $\boldsymbol{V} = \boldsymbol{x}\boldsymbol{W}_v$, where $\boldsymbol{W}_q$, $\boldsymbol{W}_k$ and $\boldsymbol{W}_v$ are learnable projection weights.

## 2.2 Unbinding tuners from foundation models

For the adaptation of the foundation models to downstream applications, the existing PETL approaches mostly resort to adjusting the output of MHA, FFN, or the Transformer block composed of MHA and FFN in various ways. We choose popular and exemplary approaches and unbind their structures from foundation models. Here, we provide the unbinding formulations of prefix tuning [41] and prompt tuning [28] for MHA adaptation, as well as adapter tuning [25] for FFN adaptation.

**Prefix tuning** [41] prepends learnable parameters, *i.e.,* prefix tokens, to the projected keys and values:

$$\text{MHA}_{\text{pre}} = \text{Attn}(\boldsymbol{x}\boldsymbol{W}_q, [\boldsymbol{K}_{pre}; \boldsymbol{x}\boldsymbol{W}_k], [\boldsymbol{V}_{pre}; \boldsymbol{x}\boldsymbol{W}_v]), \tag{2}$$

where $\boldsymbol{K}_{pre}$ and $\boldsymbol{V}_{pre}$ are prefix tokens. Essentially, if we view this as performing MHA separately between $(\boldsymbol{Q}, \boldsymbol{K}, \boldsymbol{V})$ and between $(\boldsymbol{Q}, \boldsymbol{K}_{pre}, \boldsymbol{V}_{pre})$, we unbind prefix tuning as follows:

$$\text{MHA}_{\text{pre}} = (1 - \lambda)\underbrace{\text{Attn}(\boldsymbol{Q}, \boldsymbol{K}, \boldsymbol{V})}_{\text{original attention}} + \lambda\underbrace{\text{Attn}(\boldsymbol{Q}, \boldsymbol{K}_{pre}, \boldsymbol{V}_{pre})}_{\text{prefix attention in parallel}}, \tag{3}$$

where $\lambda$ weighs between the original and prefix attention. Detailed value for $\lambda$ and the derivation process are included in appendix A. In this way, the original MHA in the foundation model $\text{Attn}(\boldsymbol{Q}, \boldsymbol{K}, \boldsymbol{V})$ and the prefix attention $\text{Attn}(\boldsymbol{Q}, \boldsymbol{K}_{pre}, \boldsymbol{V}_{pre})$ can be computed independently. The unbinding formulation of prefix tuning can be seen in Fig. 2a.

**Prompt tuning** [28] appends latent tokens to the input token before performing MHA in the backbone:

$$\text{MHA}_{\text{pro}} = \text{Attn}([\boldsymbol{x}; \boldsymbol{x}_{pro}]\boldsymbol{W}_q, [\boldsymbol{x}; \boldsymbol{x}_{pro}]\boldsymbol{W}_k, [\boldsymbol{x}; \boldsymbol{x}_{pro}]\boldsymbol{W}_v), \tag{4}$$

where $\boldsymbol{x}_{pro}$ are prompt tokens concatenated to the input token $\boldsymbol{x}$ in the first layer or between multiple layers. Similar to prefix tuning, the unbinding formulation of prompt tuning is as follows:

$$\text{MHA}_{\text{pro}} = [(1 - \lambda)\underbrace{\text{Attn}(\boldsymbol{Q}, \boldsymbol{K}, \boldsymbol{V})}_{\text{original attention}} + \lambda\underbrace{\text{Attn}(\boldsymbol{Q}, \boldsymbol{K}_{pro}, \boldsymbol{V}_{pro})}_{\text{prompt attention in parallel}}; \boldsymbol{D}], \tag{5}$$

where $\boldsymbol{K}_{pro} = \boldsymbol{x}_{pro}\boldsymbol{W}_k$ and $\boldsymbol{V}_{pro} = \boldsymbol{x}_{pro}\boldsymbol{W}_v$. $\boldsymbol{D}$ denotes disposable parts that would not affect the output of MHA, where $\boldsymbol{D} = (1 - \beta)\text{Attn}(\boldsymbol{Q}_{pro}, \boldsymbol{K}_{pro}, \boldsymbol{V}_{pro}) + \beta\text{Attn}(\boldsymbol{Q}_{pro}, \boldsymbol{K}, \boldsymbol{V})$. $\lambda$ and $\beta$ are individual weights. More details can be seen in appendix A. The unbinding formulation of prompt tuning can be seen in Fig. 2b.

**Adapter tuning** [25] typically inserts a multi-layer perceptron (MLP) after FFN. Since the MLP could be performed independently, we simply re-route the adapter and connect it in parallel to the FFN as in Fig. 2c. The resultant unbinding formulation of the adapters is as follows:

$$\text{FFN}_{\text{adapter}} = \underbrace{\text{FFN}(\boldsymbol{x})}_{\text{original module}} + \underbrace{\phi(\text{FFN}(\boldsymbol{x})\boldsymbol{W}_{down})\boldsymbol{W}_{up}}_{\text{adapter module in parallel}}, \tag{6}$$

where $\boldsymbol{W}_{down}$ and $\boldsymbol{W}_{up}$ denote the weights for the down- and up-projection layers, respectively.

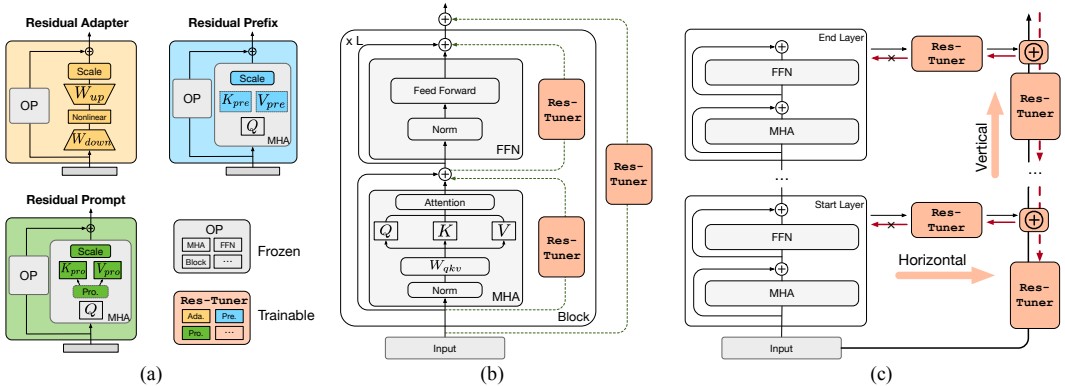

Figure 3: **Structure illustration** of (a) various `Res-Tuners` in our unbinding form, (b) our `Res-Tuning` framework that is flexible and efficient, and (c) `Res-Tuning-Bypass`, the memory-efficient version of our framework.

Table 1: **Empirical equivalence** of PETL methods and their counterparts in our unbinding form.

| Method | ViT/B-16 (IN-21K) | | | ViT/L-14 (CLIP) | | |
| --- | --- | --- | --- | --- | --- | --- |
| | Original | Unbinding form | $\Delta$ | Original | Unbinding form | $\Delta$ |
| Adapter [25] | 92.34 | 92.34 | 0.00 | 92.43 | 92.44 | +0.01 |
| Prefix [41] | 91.90 | 91.88 | -0.02 | 90.99 | 91.01 | +0.02 |
| Prompt [28] | 92.21 | 92.24 | +0.03 | 91.86 | 91.83 | -0.03 |

### 2.3 Empirical equivalency of the unbinding formulation

After we have obtained the unbinding formulation of popular PETL methods with theoretical derivation, Tab. 1 shows empirical evidence of the equivalency between the original formulation and our unbinding formulation. We carry out the comparison between two formulations on CIFAR-100, using ViT [13] pre-trained on ImageNet-21K and by CLIP [59]. For all cases, we observe that the performance discrepancy between the original and our unbinding form is within a reasonable range ($\pm$ 0.03), which shows that our formulation encompasses existing PETL methods effortlessly.

## 3 Res-Tuning

### 3.1 Unified formulation of Res-Tuning

Given the unbinding formulation of the existing PETL methods, we can derive a unified formulation as the combination of the frozen pre-trained operation and the tuner with learnable parameters:

$$\boldsymbol{x}' = \text{OP}(\boldsymbol{x}) + \texttt{Res-Tuner}(\boldsymbol{x}), \tag{7}$$

where OP denotes existing operations in the pre-trained backbone such as MHA and FFN, while the `Res-Tuner` represents the learnable structures that are connected in parallel to the existing operations.

With this unified unbinding formulation, the design of the tuner structure can now be independent of the original operation in the base model. This leads to unprecedented flexibility in parameter-efficient tuning, which enables us to explore various instantiations of the tuner (as in Fig. 3a) for the base structures. For example, we found in Tab. 2a that, instantiating the `Res-Tuner` with prompt tuning for FFN results in a better performance compared to adapters.

Further, the structural disentanglement also allows for the flexible combination of various tuning strategies. In Fig. 3b, we instantiate our `Res-Tuning` framework by associating one `Res-Tuner` with every operation in the base model, including MHA, FFN, and the whole Transformer block.

### 3.2 Res-Tuning-Bypass: Towards memory-efficient PETL

The unified formulation of `Res-Tuning` provides a viable solution for flexible and parameter-efficient transfer learning. However, since it is directly derived from existing PETL methods, it shares the

same vulnerability as existing PETL solutions. Despite the parameter efficiency, they require back-propagation through the massive parameters in the pre-trained backbone, which leads to unnecessary extra consumption of memory and computation.

To avoid this, we further present a memory-efficient version of our `Res-Tuning` framework, dubbed as `Res-Tuning-Bypass`, as in Fig. 3c. Specifically, we remove the data flow from the `Res-Tuner` to the backbone, such that the `Res-Tuner` is detached from the pre-trained architectures. In this way, we form a bypass network constructed by `Res-Tuners` in parallel with the backbone. Formally, given the tokenized feature $x_0$ and the output feature of the $l$-th layer $x_l$ from the pre-trained model, our bypass network is formulated as:

$$
\begin{aligned}
x_0^{\text{bypass}} &= x_0, \\
x_l^{\text{bypass}} &= \lambda \texttt{Res-Tuner}(x_l) + (1 - \lambda)\texttt{Res-Tuner}(x_{l-1}^{\text{bypass}}), l \geq 1,
\end{aligned}
\tag{8}
$$

where $\lambda$ is a learnable parameter followed by a sigmoid function, which is initialized to 0.5. As demonstrated in Fig. 3c, we group the `Res-Tuners` in the bypass network into horizontal ones and vertical ones, respectively processing the output feature of the $l$-th layer from the backbone and the $(l-1)$-th feature from the bypass network. Within each group, we keep the structure identical. Thanks to the flexibility of our unbinding formulation, we can also explore different instantiations of the `Res-Tuner` and various combinations of the existing tuners.

## 4 Empirical evaluations on discriminative tasks

### 4.1 Experimental setups

**Evaluation scenarios.** We mainly analyze the flexibility and efficiency of our proposed `Res-Tuning` framework and evaluate the discriminative capabilities on three different scenarios: *transfer learning*, *few-shot learning*, and *domain generalization*.

**Baselines.** Apart from the traditional downstream adaptation methods like fully fine-tuning and linear probing, we divide existing tuning approaches into two categories: **(i)** methods focusing on parameter-efficiency, including adapter tuning [25], prefix tuning [41], VPT [28], LoRA [27], AdaptFormer [7], SSF [42], and NOAH[83]; **(ii)** methods focusing on memory-efficiency, including Side-Tuning [82], and LST [68]. Since these two categories have distinct characteristics with respect to the parameters, memory, and performance, we mainly compare our `Res-Tuning` framework within the former category, while the `Res-Tuning-Bypass` is mainly compared in the latter one.

**Implementation details.** For most experiments, we adopt ViT-B/16 [13] pre-trained on ImageNet-21K [11] as the backbone model, following VPT [28]. Unless otherwise specified, the middle of the adapter, as well as the number of prefix and prompt tokens in our `Res-Tuning` are set to 10 for parameter efficiency. We include the training details in appendix C. For all the tasks, we use top-1 accuracy as the main evaluation metric.

### 4.2 Analysis on flexibility

**Flexible combination between backbone structures and tuners.** Since our unbinding formulation allows for the structural disentanglement between the frozen structure in the backbone and the tuner with learnable parameters, it enables us to explore various combinations of the operation and the tuners. Here, we experiment with various instantiations of OP and `Res-Tuner` in our `Res-Tuning` framework, and the number of tuners is limited to one tuner per block (`Single-Res-Tuner`). The results are presented in Tab. 2a. We found that the default combination in existing approaches (prefix and prompt for MHA adaptation and adapter for FFN adaptation) is far from optimal, and connecting prompt tuning to FFN results in the best performance.

**Flexible combination of multiple tuning strategies.** Next, we show that the flexibility brought by our unbinding formulation could also effortlessly lead to stronger tuning strategies by combining various tuners in our unbinding formulation. Here, we consider two tuners per block (`Dual-Res-Tuner`), respectively connected to MHA and FFN. As in Tab. 2b, employing two tuners for each block brings notable improvements over the `Single-Res-Tuner` variants, with employing two adapters respectively for MHA and FFN achieving the strongest performance of 93.25. On top of the best performing `Dual-Res-Tuner` model, we further attach tuners at the block level in Tab. 2c. With

Table 2: **Exploration of various combinations** of operations in the pre-trained backbone and various `Res-Tuners` achieves a stronger performance compared to the existing tuning strategies on CIFAR-100. Adapter, prefix, and prompt are abbeviated as Ada., Pre. and Pro., respectively.

(a) `Single-Res-Tuner`.

| Tuner\OP | MHA | FFN | Block |
|---|---|---|---|
| Res-Ada. | 92.46 | 92.34 | 92.49 |
| Res-Pre. | 91.88 | 92.33 | 92.39 |
| Res-Pro. | 92.24 | **92.68** | 92.16 |

(b) `Dual-Res-Tuner`.

| MHA\FFN | Res-Ada. | Res-Pre. | Res-Pro. |
|---|---|---|---|
| Res-Ada. | **93.25** | 92.95 | 92.62 |
| Res-Pre. | 93.22 | 92.38 | 92.87 |
| Res-Pro. | 93.03 | 92.92 | 92.91 |

(c) `Tri-Res-Tuner`.

| Block | Dual-Res-Tuner |
|---|---|
| Res-Ada. | **93.28** |
| Res-Pre. | 93.20 |
| Res-Pro. | 93.16 |

Table 3: **In-depth analysis** of our `Res-Tuning` and `Res-Tuning-Bypass` in terms of performance, parameter-efficiency and memory efficiency on CIFAR-100.

(a) Parameter efficiency of `Res-Tuning` on CIFAR-100.

| Method | Full | Linear Probing | Single- | Dual- | Tri- |
|---|---|---|---|---|---|
| Acc. | 89.12 | 85.95 | 92.68 | 93.25 | 93.28 |
| Param. | 85.9M | 0.07M | 0.17M | 0.48M | 0.67M |

(b) Parameter and memory efficiency of `Res-Tuning-Bypass` on CIFAR-100.

| Method | Bypass | Res-Tuner | Acc. | Param. | Mem. |
|---|---|---|---|---|---|
| Linear probing | ✘ | - | 85.95 | 0.07M | 2.72G |
| | ✔ | None | 86.34 | 0.07M | 3.48G |
| ↓ | ✔ | Hori. | 88.08 | 0.27M | 3.66G |
| | ✔ | Vert. | 87.26 | 0.27M | 3.64G |
| Res-Tuning-Bypass | ✔ | Both | **89.33** | 0.46M | 4.72G |
| Fully fine-tuning | ✘ | - | 89.12 | 85.9M | 9.02G |

(c) Performance and efficiency comparison with existing tuning strategies on CIFAR-100.

| Method | Acc. | Param. (M) | Mem. |
|---|---|---|---|
| Full | 89.12 | 85.9 (100%) | 9.02G |
| Linear | 85.95 | 0.07 (0.08%) | 2.72G |
| *Parameter-efficient tuning methods* | | | |
| MAM-Adapter[†] [19] | 91.70 | 10.08 (11.72%) | 9.57G |
| AdaptFormer [7] | 91.86 | 1.26 (1.46%) | 6.32G |
| Res-Tuning | **93.25** | 0.48 (0.55%) | 6.85G |
| *Memory-efficient tuning methods* | | | |
| Side-Tuning [82] | 87.16 | 9.62 (11.18%) | 3.48G |
| LST[†] [68] | 88.72 | 0.93 (1.08%) | 5.26G |
| Res-Tuning-Bypass | **89.33** | 0.46 (0.53%) | 4.72G |

† denotes our own implementation since the original approach is presented for natural langauge processing.

adapters on top of the `Dual-Res-Tuner` model, we observe further slight improvements on the classification performance. Compared to existing tuning strategies of underlining in Tab. 2a, without bells and whistles, the `Tri-Res-Tuner` version of our `Res-Tuning` framework achieves at least 0.94% performance improvement.

## 4.3 Analysis on parameter-, memory- and multi-task inference-efficiency

**Parameter-efficiency.** We analyze the parameter efficiency of our `Res-Tuning` framework in Tab. 3a. Our `Single-Res-Tuner` version surpasses the performance of the fully-fine-tuned variant with less than 0.2% learnable parameters. Combining all three tuners on MHA, FFN, and block-level, we manage to outperform the fully-fine-tuned and linear evaluated variants by respectively 4.16% and 7.33%, while only using 0.67M parameters.

**Memory-efficiency.** Here, we present the evolution from linear probing to our `Res-Tuning-Bypass` in Tab. 3b. It is observed that introducing a bypass network without any tuners can help ensemble the original features obtained from different layers, thereby mildly improving the classification accuracy as compared to the linear probing approach where only the classifiers are trained. On top of that, both horizontal and vertical `Res-Tuner` bring notable performance improvement with limited parameter and memory overhead. With both horizontal and vertical `Res-Tuners` in place, our `Res-Tuning-Bypass` framework achieves stronger performance with only 52% of the memory consumption when compared with the fully-fine-tuned variant.

**Multi-task inference-efficiency.** In Fig. 4, our `Res-Tuning-Bypass` demonstrates superior multi-task inference-efficiency on both discriminative and generative tasks. For discriminative multi-task inference, we combine the validation set of 19 tasks in VTAB-1K, perform 19 tasks on every image, and obtain the overall process time for the whole validation set. For generation multi-task inference, we take one image and provide the model with 10 fixed-length prompts for generation and record the overall generation time for the 10 prompts. For existing parameter-efficient methods, the inference time grows linearly when the number of tasks grows. Compared to the fully fine-tuned variant, all existing parameter-efficient tuning strategies increase the inference time to various extents. In contrast, our `Res-Tuning-Bypass` framework significantly reduces the inference time on 19 discriminative tasks and 10 generative tasks by respectively 80.9% and 83.6% when compared with the fully-fine-tuned variant.

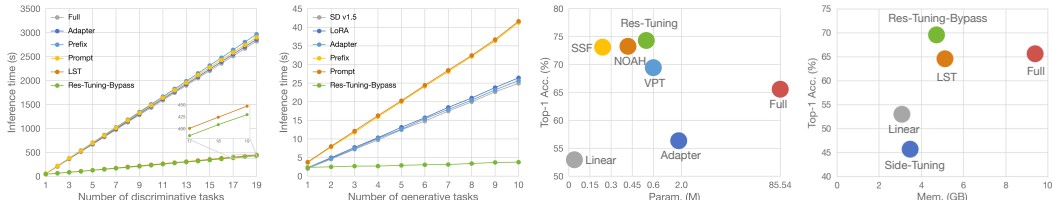

Figure 4: **Comparisons of the parameter-, memory-, and multi-task inference-efficiency**. For multi-task inference-efficiency, we evaluate `Res-Tuning-Bypass` on both discriminative and generative tasks. For parameter- and memory-efficiency, here, we show the comparisons on VTAB-1K between our approach and existing tuning strategies.

Table 4: **Performance and efficiency comparison** on the VTAB-1K benchmark with ViT-B/16 models pre-trained on ImageNet-21K. "Group Mean" denotes the average accuracy of the three subgroups. "All Mean" denotes the average accuracy of 19 downstream tasks.

| | Natural | | | | | | | Specialized | | | | Structured | | | | | | | | Group Mean | All Mean | Param. (M) | Mem. (GB) |
|---|---|---|---|---|---|---|---|---|---|---|---|---|---|---|---|---|---|---|---|---|---|---|---|
| | CIFAR-100 | Caltech101 | DTD | Flowers102 | Pets | SVHN | Sun397 | Camelyon | EuroSAT | Resisc45 | Retinopathy | Clevr-Count | Clevr-Dist | DMLab | KITTI-Dist | dSpr-Loc | dSpr-Ori | sNORB-Azim | sNORB-Elev | | | | |
| *Traditional methods* | | | | | | | | | | | | | | | | | | | | | | | |
| Full | 68.9 | 87.7 | 64.3 | 97.2 | 86.9 | 87.4 | 38.8 | 79.7 | 95.7 | 84.2 | 73.9 | 56.3 | 58.6 | 41.7 | 65.5 | 57.5 | 46.7 | 25.7 | 29.1 | 68.96 | 65.57 | 85.84 | 9.40 |
| Linear | 63.4 | 85.0 | 63.2 | 97.0 | 86.3 | 36.6 | 51.0 | 78.5 | 87.5 | 68.6 | 74.0 | 34.3 | 30.6 | 33.2 | 55.4 | 12.5 | 20.0 | 9.6 | 19.2 | 57.64 | 52.94 | 0.04 | 3.09 |
| *Parameter-efficient tuning methods* | | | | | | | | | | | | | | | | | | | | | | | |
| Adapter [25] | 74.2 | 85.7 | 62.7 | 97.8 | 87.2 | 36.4 | 50.7 | 76.9 | 89.2 | 73.5 | 71.6 | 45.2 | 41.8 | 31.1 | 56.4 | 30.4 | 24.6 | 13.2 | 22.0 | 60.52 | 56.35 | 1.82 | 6.53 |
| LoRA [27] | 67.1 | 91.4 | 69.4 | 98.8 | 90.4 | 85.3 | 54.0 | 84.9 | 95.3 | 84.4 | 73.6 | **82.9** | **69.2** | 49.8 | 78.5 | 75.7 | 47.1 | 31.0 | 44.0 | 74.60 | 72.30 | 0.29 | 6.88 |
| VPT-Deep [28] | **78.8** | 90.8 | 65.8 | 98.0 | 88.3 | 78.1 | 49.6 | 81.8 | **96.1** | 83.4 | 68.4 | 68.5 | 60.0 | 46.5 | 72.8 | 73.6 | 47.9 | **32.9** | 37.8 | 71.96 | 69.43 | 0.60 | 8.13 |
| SSF [42] | 69.0 | 92.6 | **75.1** | **99.4** | 91.8 | **90.2** | 52.9 | **87.4** | 95.9 | **87.4** | 75.5 | 75.9 | 62.3 | **53.3** | 80.6 | 77.3 | 54.9 | 29.5 | 37.9 | 75.69 | 73.10 | 0.24 | 7.47 |
| NOAH [83] | 69.6 | **92.7** | 70.2 | 99.1 | 90.4 | 86.1 | 53.7 | 84.4 | 95.4 | 83.9 | **75.8** | 82.8 | 68.9 | 49.9 | **81.7** | 81.8 | 48.3 | 32.8 | **44.2** | 75.48 | 73.25 | 0.42 | 7.27 |
| Res-Tuning | 75.2 | **92.7** | 71.9 | 99.3 | **91.9** | 86.7 | **58.5** | 86.7 | 95.6 | 85.0 | 74.6 | 80.2 | 63.6 | 50.6 | 80.2 | **85.4** | **55.7** | 31.9 | 42.0 | **76.32** | **74.10** | 0.55 | 8.95 |
| *Memory-efficient tuning methods* | | | | | | | | | | | | | | | | | | | | | | | |
| Side-Tuning [82] | 60.7 | 60.8 | 53.6 | 95.5 | 66.7 | 34.9 | 35.3 | 58.5 | 87.7 | 65.2 | 61.0 | 27.6 | 22.6 | 31.3 | 51.7 | 8.2 | 14.4 | 9.8 | 21.8 | 49.91 | 45.65 | 9.59 | 3.48 |
| LST† [68] | 58.0 | 87.1 | 66.2 | 99.1 | 89.7 | 63.2 | 52.6 | 81.9 | 92.2 | 78.5 | 69.4 | 68.6 | 56.1 | 38.8 | **73.4** | 72.9 | 30.5 | 16.6 | 31.0 | 67.56 | 64.52 | 0.89 | 5.13 |
| Res-Tuning-Bypass | 64.5 | 88.8 | 73.2 | 99.4 | 90.6 | 63.5 | 57.2 | 85.5 | 95.2 | 82.4 | 75.2 | 70.4 | 61.0 | 40.2 | 66.8 | 79.2 | 52.6 | 26.0 | 49.3 | 72.32 | 69.51 | 0.42 | 4.73 |

† denotes our own implementation since the original approach is proposed for natural language processing.

## 4.4 Comparisons with existing tuning strategies on different scenarios

**Transfer Learning.** We mainly evaluate our approach on the basic transfer learning scenario, where pre-trained models are fine-tuned on different downstream tasks. CIFAR-100 [35] is a standard general-purpose image classification dataset. VTAB-1K [80] is composed of 19 various visual classification tasks falling into three categorizations, *i.e.,* natural, specialized, and structured. We compare the performance of our approach and other baseline methods on the following:

*CIFAR-100.* We present the comparisons to existing tuning strategies in Tab. 3c. For parameter-efficient methods, our `Res-Tuning` framework notably improves over AdaptFormer [7] by 1.39%, using only 0.55% extra parameters, which is 0.91% less than AdaptFormer [7]. For memory-efficient methods, we outperform LST [68] by 0.61% with memory reduced by 0.54G (10%).

*VTAB-1K.* In Tab. 4, we present comprehensive evaluation on the 19 datasets on the VTAB-1K benchmark. Both our `Res-Tuning` and `Res-Tuning-Bypass` outperform existing approaches respectively within the group of parameter- and memory-efficient tuning methods. Specifically, our `Res-Tuning` framework achieves a 0.85% improvement in terms of the average performance on 19 datasets, compared to the previous best performance. For our `Res-Tuning-Bypass`, we outperform LST [68] in 18 out of 19 tasks and overall by 4.99% in terms of average performance. This is achieved with even fewer parameters and memory consumption. We further visualize the parameter vs. performance curve and memory vs. performance curve in Fig. 4 to show the advantage of our `Res-Tuning` and `Res-Tuning-Bypass` framework on VTAB-1K.

**Few-Shot Learning.** To evaluate the ability of our approach to adapt with only a few training samples, we follow the few-shot evaluation protocol in NOAH [83], using {1, 2, 4, 8, 16}-shots for training and full test data. We conduct experiments on five fine-grained datasets, including FGVC-Aircraft [51], Food-101 [4], Oxford Flowers [55], Oxford Pets [56], Stanford Cars [15].

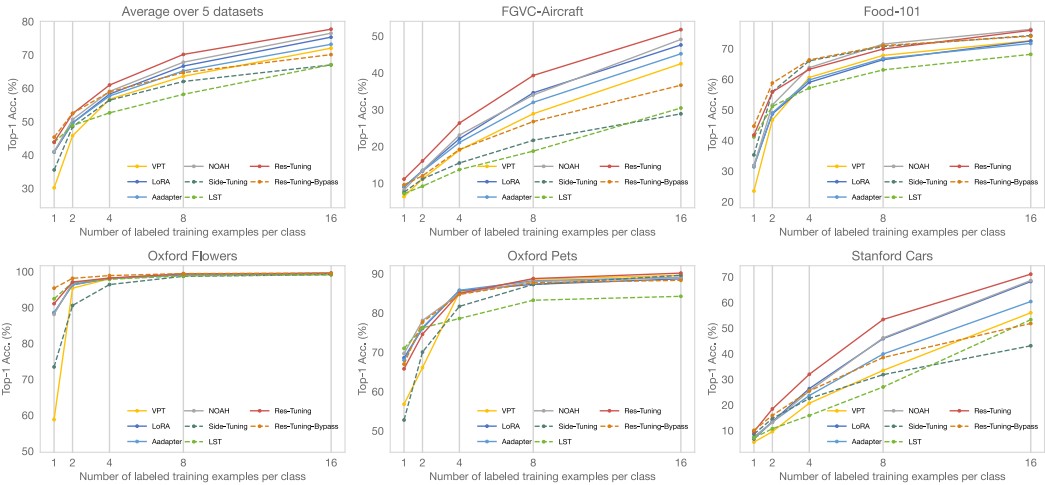

Figure 5: **Results of few-shot learning on five fine-grained visual recognition datasets.** The solid line represents the comparison of parameter-efficient tuning methods, and the dashed line represents the comparison of memory-efficient tuning methods. All results are averaged over 3 random seeds.

Table 5: **Results on domain generalization.** "Mean" denotes the average accuracy of four variants of ImageNet. All results are averaged over 3 random seeds.

| | Source | | | Target | | |
| | ImageNet | IN-V2 | IN-Sketch | IN-A | IN-R | Mean |
|---|---|---|---|---|---|---|
| *Parameter-efficient tuning methods* | | | | | | |
| Adapter [25] | 70.5 | 59.1 | 16.4 | 5.5 | 22.1 | 25.8 |
| VPT [28] | 70.5 | 58.0 | 18.3 | 4.6 | 23.2 | 26.0 |
| LoRA [27] | 70.8 | 59.3 | 20.0 | 6.9 | 23.3 | 27.4 |
| NOAH [83] | 71.5 | 66.1 | 24.8 | 11.9 | 28.5 | 32.8 |
| Res-Tuning | **78.04** | **66.58** | **29.23** | **13.15** | **29.01** | **34.50** |
| *Memory-efficient tuning methods* | | | | | | |
| Side-Tuning [82] | 74.57 | 62.52 | 23.55 | 10.37 | 25.06 | 30.38 |
| LST [68] | 70.00 | 57.04 | 14.39 | 7.21 | 17.02 | 23.92 |
| Res-Tuning-Bypass | **77.30** | **65.23** | **27.39** | **10.66** | **26.45** | **32.43** |

As the results are shown in Fig. 5, in terms of the overall performance (top-left), both `Res-Tuning` and `Res-Tuning-Bypass` demonstrate clear advantages over other corresponding parameter-efficient and memory-efficient tuning strategies of few-shot learning on five FGVC datasets. We also observe that `Res-Tuning-Bypass` performs as well as or even better than the non-memory-efficient methods on one or two shots with low training samples on serveral datasets.

**Domain Generalization.** To evaluate the robustness of our approach to distribution shift, we train a model on the source domain using 16 shots per category and test it on both the source and target domain. The source domain uses ImageNet-1K [11]) and the target domains use four other variants of ImageNet, including ImageNet-V2 [62], ImageNet-Sketch [73], ImageNet-A [24], ImageNet-R [23]. The results in Tab. 5 prove that our approach is robust under domain shift. Our `Res-Tuning` goes beyond NOAH [83] by 6.54% on ImageNet and 1.7% on the average accuracy of four variants of ImageNet. Furthermore, `Res-Tuning-Bypass` also demonstrates stronger robustness than the memory-efficient baselines and outperforms most existing parameter-efficient tuning methods.

## 5 Empirical evaluations on generative task

### 5.1 Experimental setups

**Downstream tasks.** To provide a more comprehensive evaluation of our `Res-Tuning` framework, we further apply it to the text-to-image generation task. Following [63], we evaluate the text-to-image

Table 6: **Comparison of FID and efficiency on COCO2017.** Following the default settings of stable diffusion[1], we sample 10k captions from the validation set for generating images of size $512^2$ using 50 PLMS steps with classifier-free guidance scale 3.0 and compare against the full validation set.

| Method | FID | Param. (M) | Mem. (GB) | Train (Hour/Epoch) |
|---|---|---|---|---|
| SD v1.5 | 15.48 | - | - | - |
| + Full | 14.85 | 862 (100%) | 72.77 | 1.98 |
| + LoRA | 14.50 | 9.96 (1.15%) | 61.03 | 1.42 |
| + Adapter | 14.73 | 2.51 (0.29%) | 54.30 | 1.30 |
| + Prefix | 15.36 | 4.99 (0.58%) | 64.91 | 2.20 |
| + Prompt | 14.90 | **1.25** (0.14%) | 63.70 | 2.17 |
| + Res-Tuning | **13.96** | 2.54 (0.29%) | 54.49 | 1.38 |
| + Res-Tuning Bypass | 14.89 | 3.76 (0.44%) | **21.35** | **0.82** |

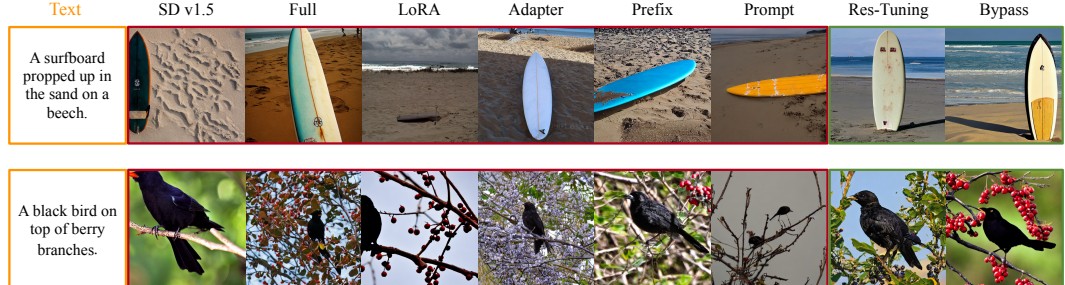

Figure 6: **Qualitative results** of SD v1.5, existing tuning strategies and our `Res-Tuning` on COCO2017 validation set [43]. We frame our results in green and others in red.

generation performance on COCO2017 dataset [43]. The main quantitative evaluation metric is the FID score and we also perform instance-level fine-tuning transfer on the fine-grained datasets [4, 55].

**Baselines.** We experiment with the version 1.5 of stable diffusion [63] (SD) model. On COCO2017, we compare our `Res-Tuning` framework with zero-shot SD, SD with fully fine-tuning as well as SD with other existing tuning methods. On the fine-grained datasets, we employ DreamBooth [65] as our baseline and employ various tuning methods including our own for comparison.

## 5.2 Main results

**Text-to-image generation on COCO.** We show the comparison between our `Res-Tuning` framework and other approaches both quantitatively and qualitatively. The quantitative comparison is presented in Tab. 6. Compared with the fully fine-tuned baseline, our `Res-Tuning` framework improves the FID score by 0.89, using only 0.29% of the parameters. It is noteworthy that our `Res-Tuning` framework is the only approach that reaches a FID score below 14, while the best existing tuning strategy on the task is LoRA [27] achieving 14.50 with 4x the number of parameters in `Res-Tuning`. Hence, employing `Res-Tuning` could greatly facilitate the adaptation of pre-trained text-to-image generation models to downstream tasks.

A highlight is observed that our `Res-Tuning-Bypass` framework reduces the memory consumption for tuning the SD v1.5 model to only 29% while maintaining a similar performance (14.89 vs 14.85). Meanwhile, the time consumption for training the model is reduced to 41%. In terms of the reduction in the memory and time consumption, our `Res-Tuning-Bypass` framework outperforms the best tuning strategy by 3.3x (70.7% memory reduction of `Res-Tuning-Bypass` vs. 25.3% that of adapter) and 1.7x (58.6% reduction in time consumption of `Res-Tuning-Bypass` vs. 34.3% that of adapter), respectively.

The qualitative results are presented in Fig. 6. Both `Res-Tuning` and `Res-Tuning-Bypass` show a better alignment between the text and the generated image, where the surfboard is propped up in our generated images. `Res-Tuning` also demonstrates a better fidelity where the feather texture is realistic in the generated black bird.

---

[1]https://huggingface.co/runwayml/stable-diffusion-v1-5

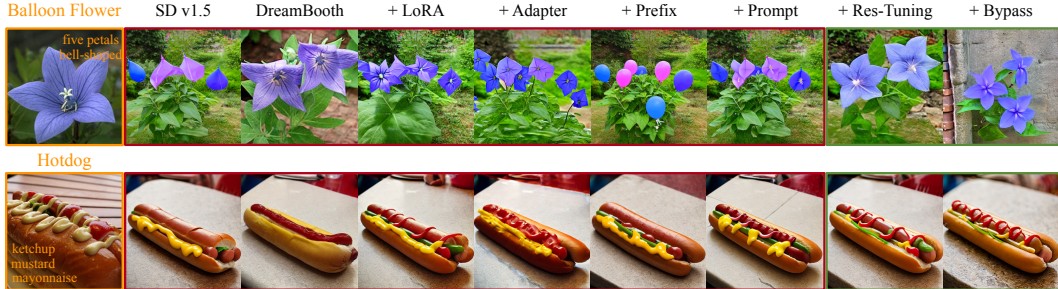

Figure 7: **Qualitative results** of SD v1.5, DreamBooth, existing tuning strategies, and our `Res-Tuning` on Oxford Flowers and Food-101 fine-grained dataset with the same generated seed. We frame our results in green and others in red.

**Text-to-image generation on Oxford Flowers and Food-101.** Here, we evaluate the transfer ability of existing tuning strategies on specific fine-grained categories. During every evaluation process, we select data from one specific category to train the model. The qualitative results are demonstrated in Fig. 7. It is observed that `Res-Tuning` presents a better view in terms of fidelity and the correct understanding of ambiguous categories. For example, the balloon flower is bell-shaped and has five petals, while existing approaches generate flowers with the wrong number of petals or even balloons rather than real flowers. Instead, both our `Res-Tuning` and our `Res-Tuning-Bypass` retain correct semantics and fine-grained features.

## 6 Related work

**Transformers in computer vision.** Transformers [70] have demonstrated strong capabilities in various fields [5, 61, 12, 59, 2, 17, 48, 53]. In computer vision, Vision Transformers (ViT) [13] are widely applied in various applications, such as visual classification [48, 39], object detection [67, 6], segmentation [84, 74] and generation [63, 57]. Owing to the strong scalability of the Transformer backbone [31, 10], recent endeavors either focus on expanding the size of the ViT [81] or training on a larger corpus of data in an unsupervised manner [75, 14].

**Parameter-efficient tuning.** Despite the strong performance and generalization ability brought by scaling up the Transformers, it also makes the adaptation to the downstream tasks expensive and almost infeasible. Hence, parameter-efficient transfer learning (PETL) emerged [26, 25, 76, 41, 85, 28]. Existing PETL methods could be generally categorized into three types: (i) MHA-based tuning embeds tunable parameters in the multi-head self-attention layers [27, 76, 28, 41, 45, 46]. (ii) FFN-based tuning methods represented by adapters [25] and its generalized versions [58, 33, 32, 19] introduce a multi-layer perceptron to the FFN layer. (iii) Other tuning methods adapt certain existing parameters [79, 42]. Closely related to our work, some recent efforts are devoted to finding out the optimal design paradigm of the tuning modules by neural architecture search [83]. Instead, we show that through our `Res-Tuning` framework, a stronger tuning strategy can be easily found by the flexible combination of several existing tuning strategies in our unbinding form.

**Memory-efficient tuning.** Since the structures of the existing PETL methods are deeply embedded in the backbone structure, back-propagation is required through the massive parameters of the pre-trained models, leading to unnecessary extra consumption of memory and computation. Hence, Side-Tuning [82] and LST [68] in natural language processing connect a side network in parallel with the pre-trained model to avoid data flow from the trainable parameters to the frozen ones. Our `Res-Tuning-Bypass` is inspired by the conceptual idea of these approaches. Compared to Side-Tuning and LST, we show that our `Res-Tuning-Bypass` is more flexible and memory-efficient.

## 7 Conclusion

In this work, we unbind the tuners from the backbone and form a flexible and efficient tuning paradigm `Res-Tuning`. With `Res-Tuning`, we are able to find stronger tuning strategies compared to existing ones. On top of `Res-Tuning`, we further extend a memory-efficient `Res-Tuning-Bypass`, which significantly reduces the memory consumption and multi-task inference cost. We hope our discoveries can facilitate further research in the flexible and efficient tuning of large foundation models.

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

In the appendix, we provide a detailed derivation process for turning existing PETL methods into our unbinding formulation (appendix A), demonstration of our unbinding formulation as a unified formulation for existing PETL methods (appendix B), more implementation details (appendix C) including the dataset used, architectures for discriminative and generative tasks, and hyperparameters used in training, additional results on discriminative tasks (appendix D) and generative tasks (appendix E).

## A   Detailed derivations

In this section, we provide the detailed deriving process for the existing prefix [41] and prompt [28] tuning to be turned into our unbinding formulation.

**Prefix tuning** [41]: The following is the detailed derivation of Eq. (3).

$$
\begin{aligned}
\mathrm{MHA}_{\mathrm{pre}} &= \mathrm{Attn}(\boldsymbol{x}\boldsymbol{W}_q, [\boldsymbol{K}_{pre}; \boldsymbol{x}\boldsymbol{W}_k], [\boldsymbol{V}_{pre}; \boldsymbol{x}\boldsymbol{W}_v]) \\
&= \mathrm{softmax}\left(\boldsymbol{x}\boldsymbol{W}_q[\boldsymbol{K}_{pre}; \boldsymbol{x}\boldsymbol{W}_k]^{\top}\right) \begin{bmatrix} \boldsymbol{V}_{pre} \\ \boldsymbol{x}\boldsymbol{W}_v \end{bmatrix} \\
&= (1 - \lambda(\boldsymbol{x}))\,\mathrm{softmax}\left(\boldsymbol{x}\boldsymbol{W}_q\boldsymbol{W}_k^{\top}\boldsymbol{x}^{\top}\right)\boldsymbol{x}\boldsymbol{W}_v + \lambda(\boldsymbol{x})\,\mathrm{softmax}\left(x\boldsymbol{W}_q\boldsymbol{K}_{pre}^{\top}\right)\boldsymbol{V}_{pre} \\
&= (1 - \lambda(\boldsymbol{x}))\,\mathrm{Attn}\left(\boldsymbol{x}\boldsymbol{W}_q, \boldsymbol{x}\boldsymbol{W}_k, \boldsymbol{x}\boldsymbol{W}_v\right) + \lambda(\boldsymbol{x})\,\mathrm{Attn}\left(\boldsymbol{x}\boldsymbol{W}_q, \boldsymbol{K}_{pre}, \boldsymbol{V}_{pre}\right) \\
&= (1 - \lambda(\boldsymbol{Q}, \boldsymbol{K}, \boldsymbol{K}_{pre}))\underbrace{\mathrm{Attn}\left(\boldsymbol{Q}, \boldsymbol{K}, \boldsymbol{V}\right)}_{\text{standard attention}} + \lambda(\boldsymbol{Q}, \boldsymbol{K}, \boldsymbol{K}_{pre})\underbrace{\mathrm{Attn}\left(\boldsymbol{Q}, \boldsymbol{K}_{pre}, \boldsymbol{V}_{pre}\right)}_{\text{independent of } \boldsymbol{K}_{pre}, \boldsymbol{V}_{pre}}
\end{aligned}
\tag{9}
$$

where $\boldsymbol{Q}$, $\boldsymbol{K}$ denote the original query and original key, $\boldsymbol{K}_{pre}$ and $\boldsymbol{V}_{pre}$ are prefix key tokens and prefix value tokens, and

$$
\lambda(\boldsymbol{Q}, \boldsymbol{K}, \boldsymbol{K}_{pre}) = \frac{\sum_i \exp\left(\boldsymbol{Q}\boldsymbol{K}_{pre}^{\top}\right)_i}{\sum_i \exp\left(\boldsymbol{Q}\boldsymbol{K}^{\top}\right)_i + \sum_j \exp\left(\boldsymbol{Q}\boldsymbol{K}_{pre}^{\top}\right)_j},
\tag{10}
$$

**Prompt tuning** [28]: The following is the detailed derivation of Eq. (5).

$$
\begin{aligned}
\mathrm{MHA}_{\mathrm{pro}} &= \mathrm{Attn}\left([\boldsymbol{x}; \boldsymbol{x}_{pro}]\boldsymbol{W}_q, [\boldsymbol{x}; \boldsymbol{x}_{pro}]\boldsymbol{W}_k, [\boldsymbol{x}; \boldsymbol{x}_{pro}]\boldsymbol{W}_v\right) \\
&= \mathrm{concat}\left(\mathrm{softmax}\left(\boldsymbol{x}\boldsymbol{W}_q[\boldsymbol{x}\boldsymbol{W}_k; \boldsymbol{x}_{pro}\boldsymbol{W}_k]^{\top}\right)\begin{bmatrix} \boldsymbol{x}\boldsymbol{W}_v \\ \boldsymbol{x}_{pro}\boldsymbol{W}_v \end{bmatrix}, \right. \\
&\qquad \left. \mathrm{softmax}\left(\boldsymbol{x}_{pro}\boldsymbol{W}_q[\boldsymbol{x}\boldsymbol{W}_k; \boldsymbol{x}_{pro}\boldsymbol{W}_k]^{\top}\right)\begin{bmatrix} \boldsymbol{x}\boldsymbol{W}_v \\ \boldsymbol{x}_{pro}\boldsymbol{W}_v \end{bmatrix}\right) \\
&= \mathrm{concat}\left((1 - \lambda(\boldsymbol{Q}, \boldsymbol{K}, \boldsymbol{K}_{pro}))\,\mathrm{Attn}\left(\boldsymbol{Q}, \boldsymbol{K}, \boldsymbol{V}\right) + \lambda(\boldsymbol{Q}, \boldsymbol{K}, \boldsymbol{K}_{pro})\,\mathrm{Attn}\left(\boldsymbol{Q}, \boldsymbol{K}_{pro}, \boldsymbol{V}_{pro}\right), \right. \\
&\qquad (1 - \beta(\boldsymbol{Q}_{pro}, \boldsymbol{K}_{pro}, \boldsymbol{K}))\,\mathrm{Attn}\left(\boldsymbol{Q}_{pro}, \boldsymbol{K}_{pro}, \boldsymbol{V}_{pro}\right) \\
&\qquad \left. + \beta(\boldsymbol{Q}_{pro}, \boldsymbol{K}_{pro}, \boldsymbol{K})\,\mathrm{Attn}\left(\boldsymbol{Q}_{pro}, \boldsymbol{K}, \boldsymbol{V}\right)\right)
\end{aligned}
\tag{11}
$$

where $\boldsymbol{K}_{pro}$ and $\boldsymbol{V}_{pro}$ are prompt key tokens and prompt value tokens, and

$$
\lambda(\boldsymbol{Q}, \boldsymbol{K}, \boldsymbol{K}_{pro}) = \frac{\sum_i \exp\left(\boldsymbol{Q}\boldsymbol{K}_{pro}^{\top}\right)_i}{\sum_i \exp\left(\boldsymbol{Q}\boldsymbol{K}^{\top}\right)_i + \sum_j \exp\left(\boldsymbol{Q}\boldsymbol{K}_{pro}^{\top}\right)_j},
\tag{12}
$$

$$
\beta(\boldsymbol{Q}_{pro}, \boldsymbol{K}_{pro}, \boldsymbol{K}) = \frac{\sum_i \exp\left(\boldsymbol{Q}_{pro}\boldsymbol{K}^{\top}\right)_i}{\sum_i \exp\left(\boldsymbol{Q}_{pro}\boldsymbol{K}_{pro}^{\top}\right)_i + \sum_j \exp\left(\boldsymbol{Q}_{pro}\boldsymbol{K}^{\top}\right)_j},
\tag{13}
$$

# B  Unified formulation

From Eq. (7), we derive a unified formulation for existing PETL methods, which is the combination of the frozen pre-trained operation (OP) and the tuner with learnable parameters (`Res-Tuner`). The following is the detailed instantiation of this unified formulation (as in Tab. 7). Thanks to the form of unbinding, tuners can be treated as individual and flexibly combined. Our framework can effectively encompass existing tuning methods in a residual form and is not limited to the OP and `Res-Tuner` mentioned in our work. For example, for MAM-Adapter, we are free to attach Res-Prefix and Res-Adapter to MHA and FFN modules, respectively. In particular, when new PETL methods are proposed, they can be quickly applied (abbreviated as New-Tuning), and different methods can be arbitrarily combined (abbreviated as Mix-Tuning).

Table 7: The combination of `Res-Tuning`.

| Method | OP | `Res-Tuner` |
|---|---|---|
| Adapter [25] | FFN | Res-Adapter |
| Prefix [41] | MHA | Res-Prefix |
| Prompt [28] | MHA | Res-Prompt |
| LoRA [27] | Weight | MLP |
| BitFit [79] | Bias | Parameter |
| AdaptFormer [7] | FFN | Res-Adapter |
| MAM-Adapter [19] | MHA + FFN | Res-Prefix + Res-Adapter |
| Side-Tuning [82] | All blocks | Side model |
| LST [68] | Block | ViT |
| New-Tuning | Any where | New-Tuner |
| Mix-Tuning | Any combination | Any combination |

# C  Implementation details

## C.1  Dataset description

In Tab. 8 and Tab. 9, we list the description of each dataset in our experiments, which includes the number of classes and the amount of images in training set and test set for discriminative and generative tasks, respectively.

Table 8: Datasets used for generative tasks. ⋆ denotes only part of the data is used.

| Dataset | Description | Classes | Train | | Test | |
|---|---|---|---|---|---|---|
| | | | image | prompt | image | prompt |
| *Common Objects in Context (COCO)* | | | | | | |
| COCO2017 Captions | common objects | - | 118287 | 591753 | 5000 | 25014 |
| *Fine-grained Image Generation* | | | | | | |
| Aircraft [51]⋆ | Fine-grained aircraft | 100 | 3334 | - | 3333 | - |
| Food-101 [4]⋆ | Fine-grained food | 101 | 75750 | - | 25250 | - |
| NABirds [69]⋆ | Fine-grained bird | 555 | 23929 | - | 24633 | - |
| Stanford Cars [15]⋆ | Fine-grained car | 196 | 8144 | - | 8144 | - |
| Stanford Dogs [34]⋆ | Fine-grained dog | 120 | 12000 | - | 8580 | - |
| SUN397 [78]⋆ | Fine-grained scene | 397 | 76044 | - | 16295 | - |

Table 9: Datasets used for discriminative tasks.

| Dataset | Description | Classes | Train | Test |
|---------|-------------|---------|-------|------|
| *General Image Classification* | | | | |
| CIFAR-100 [35] | General | 100 | 50000 | 10000 |
| *Fine-grained Visual Classification (FGVC)* | | | | |
| CUB-200-2011 [72] | Bird | 200 | 5994 | 5794 |
| NABirds [69] | Bird | 555 | 23929 | 24633 |
| Oxford Flowers [55] | Flower | 102 | 1020 | 6149 |
| Stanford Cars [15] | Car | 196 | 8144 | 8041 |
| Stanford Dogs [34] | Dog | 120 | 12000 | 8580 |
| *Visual Task Adaptation Benchmark (VTAB-1K)* | | | | |
| CIFAR-100 [35] | | 100 | | 10000 |
| Caltech101 [38] | | 102 | | 6084 |
| DTD [9] | | 47 | | 1880 |
| Flower102 [55] | Natural | 102 | 1000 | 6149 |
| Pets [56] | | 37 | | 3669 |
| SVHN [54] | | 10 | | 26032 |
| SUN397 [78] | | 397 | | 21750 |
| Camelyon [71] | | 2 | | 32768 |
| EuroSAT [22] | | 10 | | 5400 |
| Resisc45 [8] | Specialized | 45 | 1000 | 6300 |
| Retinopathy [18] | | 5 | | 42670 |
| Clevr-Count [30] | | 8 | | 15000 |
| Clevr-Dist [30] | | 6 | | 15000 |
| DMLab [3] | | 6 | | 22735 |
| KITTI-Dist [16] | | 4 | | 711 |
| dSpr-Loc [52] | Structured | 16 | 1000 | 73728 |
| dSpr-Ori [52] | | 16 | | 73728 |
| sNORB-Azim [36] | | 18 | | 12150 |
| sNORB-Ele [36] | | 9 | | 12150 |
| *Few-Shot Learning* | | | | |
| FGVC-Aircraft [51] | Aircraft | 100 | (1/2/4/8/16) * class | 3333 |
| Food-101 [4] | Food | 101 | (1/2/4/8/16) * class | 30300 |
| Oxford Flowers [55] | Flower | 102 | (1/2/4/8/16) * class | 2463 |
| Oxford Pets [56] | Pet | 37 | (1/2/4/8/16) * class | 3669 |
| Stanford Cars [15] | Car | 196 | (1/2/4/8/16) * class | 8041 |
| *Domain Generalization* | | | | |
| ImageNet-1K [11] | General | 1000 | 16 * class | 50000 |
| ImageNet-V2 [62] | General | 1000 | / | 10000 |
| ImageNet-Sketch [73] | General | 1000 | / | 50889 |
| ImageNet-A [24] | General | 200 | / | 7500 |
| ImageNet-R [23] | General | 200 | / | 30000 |

## C.2 Architectures design

We show the complete `Res-Tuning-Bypass` structural design according to the different task frameworks. For discriminative tasks, we design based on the architecture of Vision Transformers [13], mainly based on main blocks and unbound bypass (as in Fig. 8). For generative tasks, we design based on the stable diffusion [63] framework and innovatively apply `Res-Tuner` to U-Net [64] structure (as in Fig. 9). We attach `Res-Tuner` to the U-Net intermediate module and decoder module for more efficient training.

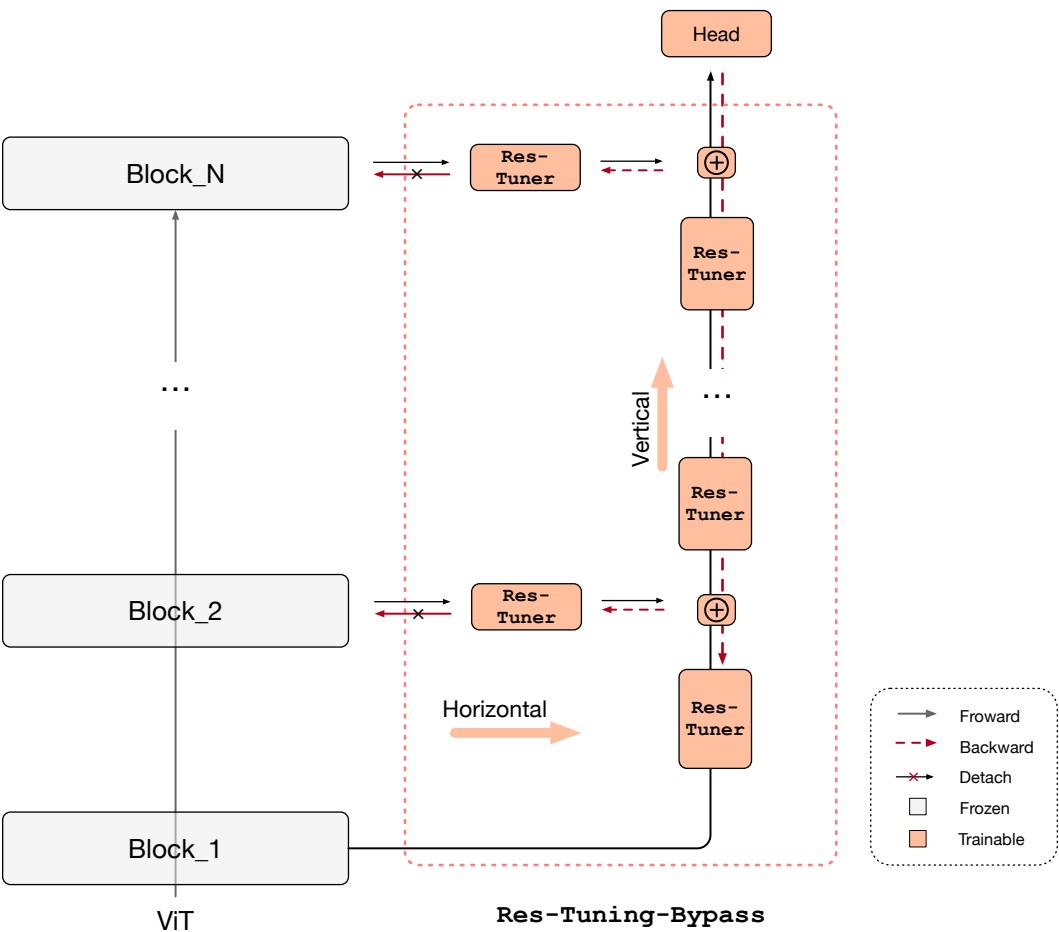

Figure 8: Architecture design for discriminative tasks.

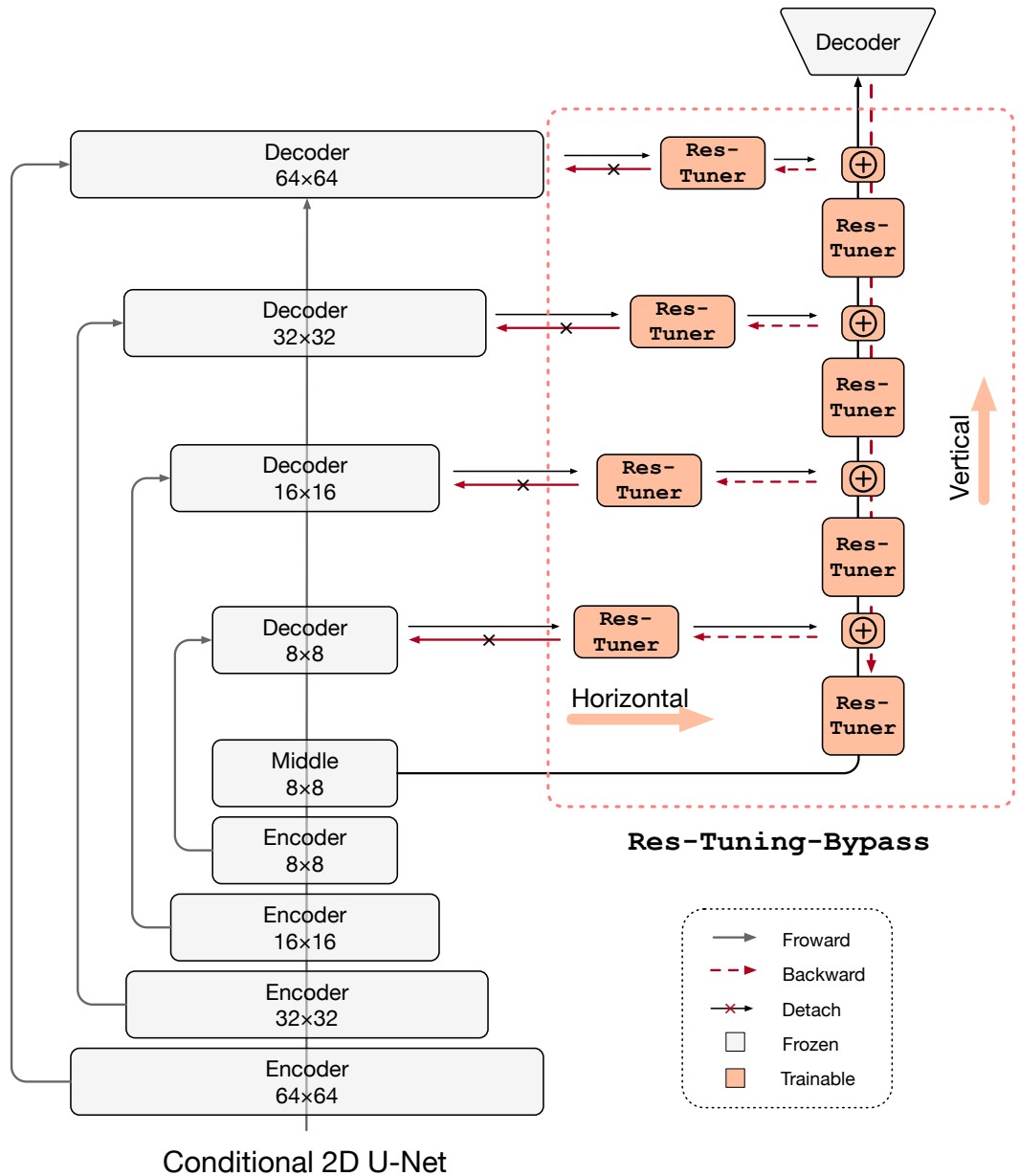

Figure 9: Architecture design for generative tasks.

## C.3  Hyperparameters

We list all the hyperparameters for discriminative (see in Tab. 10) and generative (see in Tab. 11) tasks.

Table 10: Hyperparameter selection for discriminative tasks.

| Config | Value |
|---|---|
| Batch size | 32 |
| Optimizer | AdamW [50] |
| Weight decay | 0.05 |
| Base learning rate range | {0.05, 0.01, 0.005, 0.001} |
| Learning rate schedule | Cosine decay |
| Training epochs range | {50, 100} |
| Warmup epochs | 10 |
| Augmentation | RandomResizedCrop, RandomHorizontalFlip |
| OP | MHA [70] |
| | FFN [70] |
| | Block [70] |
| Res-Tuner | Res-Adapter |
| | Res-Prefix |
| | Res-Prompt |
| Architecture | ViT/B-16 [13] |
| | ViT/L-14 [13] |
| Pre-trained | ImageNet-21K [11] |
| | CLIP [59] |
| Device | A100 × 1 |

Table 11: Hyperparameter selection for generative tasks.

| Config | Value |
|---|---|
| Batch size | 8 |
| Optimizer | AdamW [50] |
| Base learning rate | 1e-4 |
| Learning rate schedule | Constant |
| Training epochs | 10 |
| Augmentation | CenterCrop, RandomHorizontalFlip |
| Resolution | 512 × 512 |
| Sampler | PLMS [44] |
| Guidance | 3.0 |
| OP | MHA [70] |
| | FFN [70] |
| | Block [70] |
| | Conditional 2D U-Net [64] |
| Res-Tuner | Res-Adapter |
| | Res-Prefix |
| | Res-Prompt |
| Architecture | Stable Diffusion [63] |
| Pre-trained | Stable Diffusion v1.5 [63] |
| | CLIP [59] |
| Device | A100 × 8 |
| Library | Diffusers [2] |

---

[2]https://github.com/huggingface/diffusers

# D Additional experiments on discriminative tasks

## D.1 Comparisons on FGVC datasets

We compare the transfer ability of our `Res-Tuning` framework with different existing approaches for parameter- and memory-efficient transfer learning on FGVC datasets. As shown in Tab. 12, our `Res-Tuning` outperforms other tuning methods on the average accuracy of five FGVC datasets. For memory-efficient methods, `Res-Tuning-Bypass` achieves a 2.89% improvement compared to LST [68] with less memory consumption.

Table 12: **Performance and efficiency comparison** on FGVC. † denotes our own implementation.

| Datasets / Method | CUB-200-2011 | NABirds | Oxford Flowers | Stanford Cars | Stanford Dogs | Mean | Param. (M) | Mem. (GB) |
|---|---|---|---|---|---|---|---|---|
| Full | 87.3 | 82.7 | 98.8 | 84.5 | 89.4 | 88.54 | 85.98 | 9.40 |
| Linear | 85.3 | 75.9 | 97.9 | 51.3 | 86.2 | 79.32 | 0.18 | 3.09 |
| *Parameter-efficient tuning methods* | | | | | | | | |
| Adapter [25] | 87.3 | 84.3 | 98.4 | 68.4 | 88.8 | 85.46 | 1.96 | 6.55 |
| Prefix† [41] | 85.4 | 78.8 | 99.2 | 76.4 | 89.5 | 85.86 | 0.36 | 6.60 |
| VPT-Shallow [28] | 86.7 | 78.8 | 98.4 | 68.7 | 90.7 | 84.62 | 0.25 | 8.14 |
| VPT-Deep [28] | 88.5 | 84.2 | 99.0 | 83.6 | 90.2 | 89.11 | 0.85 | 8.16 |
| LoRA† [27] | 86.0 | 80.2 | 99.2 | 85.2 | 88.6 | 87.84 | 0.55 | 6.78 |
| Convpass† [29] | 86.9 | 81.4 | 99.3 | 85.7 | 89.9 | 88.66 | 0.51 | 7.44 |
| SSF [42] | 89.5 | 85.7 | **99.6** | **89.2** | 89.6 | 90.72 | 0.39 | 7.47 |
| Res-Tuning | **89.66** | **85.87** | 99.45 | 87.58 | **92.21** | **90.95** | 0.68 | 8.98 |
| *Memory-efficient tuning methods* | | | | | | | | |
| Side-Tuning [82] | 84.7 | 75.8 | 96.9 | 48.6 | 85.8 | 78.35 | 9.73 | 3.48 |
| LST† [68] | 82.98 | 76.11 | 98.83 | **78.46** | 87.89 | 84.85 | 1.04 | 5.28 |
| Res-Tuning-Bypass | **88.75** | **83.00** | **99.61** | 75.41 | **92.40** | **87.83** | 0.56 | 4.73 |

## D.2 More ablation studies

**Varying the length of dimensions.** The length of dimensions represents the hidden dimension for Res-Adapter and the number of tokens of Res-Prompt and Res-Prefix. Altering the length of dimensions affects performances, the number of parameters, and memory consumption simultaneously (see in Tab. 13). By default, we use the length that balances the number of parameters and memory, although there is a slight increase as the length grows to a certain extent.

Table 13: Ablation studies on the length of dimension for `Res-Tuning` on CIFAR-100. Default setting is marked in gray.

| Dim | Res-Tuning | | | Res-Tuning-Bypass | | |
|---|---|---|---|---|---|---|
| | Acc. | Param. | Mem. | Acc. | Param. | Mem. |
| 5 | 93.03 | 0.30M | 6.84G | 89.16 | 0.37M | 4.65G |
| 10 | **93.25** | 0.48M | 6.85G | 89.33 | 0.46M | 4.72G |
| 20 | 92.98 | 0.85M | 6.87G | 89.28 | 0.64M | 4.83G |
| 50 | 92.69 | 1.96M | 6.90G | 89.22 | 1.19M | 5.08G |
| 100 | 92.48 | 3.80M | 6.96G | **89.52** | 2.11M | 5.68G |

**Different number of block layers.** Based on the ViT/B-16 structure, we change the number of block layers in two ways: **(i)** increase from head to toe gradually (see in Tab. 14), **(ii)** discard several intermediate layers (see in Tab. 15), and evaluate the impact of using a different number of layers. It can be seen that the best performance can be achieved by using the full number of layers, but it requires the use of relatively more parameters and memory.

Table 14: Ablation studies on the tuning block layers for `Res-Tuning` on CIFAR-100. The right arrow denotes use the number of all layers from left to right. Default setting is marked in gray.

| Blocks | Res-Tuning | | | Res-Tuning-Bypass | | |
|---|---|---|---|---|---|---|
| | Acc. | Param. | Mem. | Acc. | Param. | Mem. |
| $1 \rightarrow 1$ | 92.23 | 0.11M | 6.43G | 88.50 | 0.11M | 3.58M |
| $1 \rightarrow 3$ | 92.78 | 0.18M | 6.52G | 88.60 | 0.17M | 3.73G |
| $1 \rightarrow 6$ | 92.93 | 0.28M | 6.64G | 89.05 | 0.27M | 4.09G |
| $1 \rightarrow 9$ | 93.17 | 0.38M | 6.84G | 89.32 | 0.36M | 4.53G |
| $1 \rightarrow 12$ | **93.25** | 0.48M | 6.85G | **89.33** | 0.46M | 4.72G |

Table 15: Ablation studies on the drop block layers for `Res-Tuning` on CIFAR-100. Inside the braces indicate the layer number that is discarded. The default setting is marked in gray.

| Drop block layers | Res-Tuning | | | Res-Tuning-Bypass | | |
|---|---|---|---|---|---|---|
| | Acc. | Param. | Mem. | Acc. | Param. | Mem. |
| {} | **93.25** | 0.48M | 6.85G | **89.33** | 0.46M | 4.72G |
| {3, 6, 9} | 92.88 | 0.38M | 6.73G | 89.30 | 0.36M | 4.36G |
| {2, 3, 5, 7, 9, 11} | 92.96 | 0.28M | 6.68G | 88.89 | 0.27M | 3.96G |
| {2, 3, 4, 5, 6, 8, 9, 10 ,11} | 92.71 | 0.18M | 6.56G | 88.79 | 0.17M | 3.65G |
| {1, 2, 3, 4, 5, 6, 7, 8, 9, 10, 11} | 87.77 | 0.11M | 3.31G | 88.49 | 0.11M | 3.48G |

**Different CNN backbones.** We present the results for ConvNeXt [49] and ResNet-101 [20] pre-trained on ImageNet-21K in Tab. 16. We observe that two convolutional models have different characteristics, where the performance variation of ConvNeXt is small and that of ResNet is large. In terms of the effectiveness of `Res-Tuning-Bypass`, it is observed that it outperforms the fully-finetuned version of ConvNeXt and notably improves the performance of linear probing for ResNet.

Table 16: Ablation studies on the CNN-based backbones on CIFAR-100.

| Method | ConvNext | | | ResNet-101 | | |
|---|---|---|---|---|---|---|
| | Acc. | Param. | Mem. | Acc. | Param. | Mem. |
| Full | 90.15 | 87.67 | 11.16 | 77.40 | 42.70 | 7.30 |
| Linear | 90.06 | 0.11 | 3.36 | 54.96 | 0.20 | 2.83 |
| Res-Tuning | **90.86** | 0.87 | 9.45 | **86.80** | 0.92 | 7.20 |
| Res-Tuning-Bypass | 90.51 | 1.13 | 3.63 | 72.27 | 3.63 | 4.05 |

**Experiments on NLP downstream task.** We also conduct experiments on downstream tasks beyond vision. For NLP, we perform experiments on the SST2 [66] and MNLI [77] datasets for text classification tasks. It is observed that the performance of our `Res-Tuning` framework is on par with or better than MAM-Adapter [19] in text classification, with slightly longer training time but lower memory consumption. Our `Res-Tuning-Bypass` significantly reduces the training time and memory consumption and achieves a mildly lower performance.

Table 17: Performance comparison on text classification task.

| Method | SST2 | | MNLI | | Train Param. | Test Param. | Mem. |
|---|---|---|---|---|---|---|---|
| | Acc. | Train Time | Acc. | Train Time | | | |
| MAM Adapter [19] | 94.2 | 7.2 | 87.4 | 41.4 | 46.78 (37.4%) | 0.61 (0.5%) | 22.4G |
| Res-Tuning | **94.56** | 7.9 | **87.45** | 47.3 | 0.97 (0.77%) | 0.97 (0.77%) | 19.3G |
| Res-Tuning-Bypass | 92.94 | 4.2 | 82.01 | 24.2 | 0.98 (0.78%) | 0.98 (0.78%) | 4.3G |

# E  Additional Experiments on generative tasks

## E.1  More visualizations on COCO

We present more visualization results, comparing real images, stable diffusion [63] (SD) v1.5, fully fine-tuning, LoRA [27], Res-Tuning and Res-Tuning-Bypass in Fig. 10 and more detailed generated image in Fig. 11. Specifically, the text conditions of text-to-image task are sampled from COCO Captions. To better demonstrate the advantages of our approach in understanding the concepts of text, the main elements are marked in blue, and the main modifiers or prepositions are marked in green.

## E.2  More visualizations on fine-grained datasets

We present more visualization results on fine-grained datasets (see in Fig. 12, Fig. 13, and Fig. 14), including NABirds [69], Stanford Dogs [34], Stanford Cars [15], Aircraft [51], SUN397 [78] and Food-101 [4].

# F  Limitations and Societal Impacts

This work is a tuning paradigm that is fine-tuned based on the pre-trained foundation models while freezing the backbone network, so its transfer ability depends to a large extent on the performance of the upstream model. However, when the upstream pre-training model contains illegal content training, it will also lead to the illegal use of tuning methods.

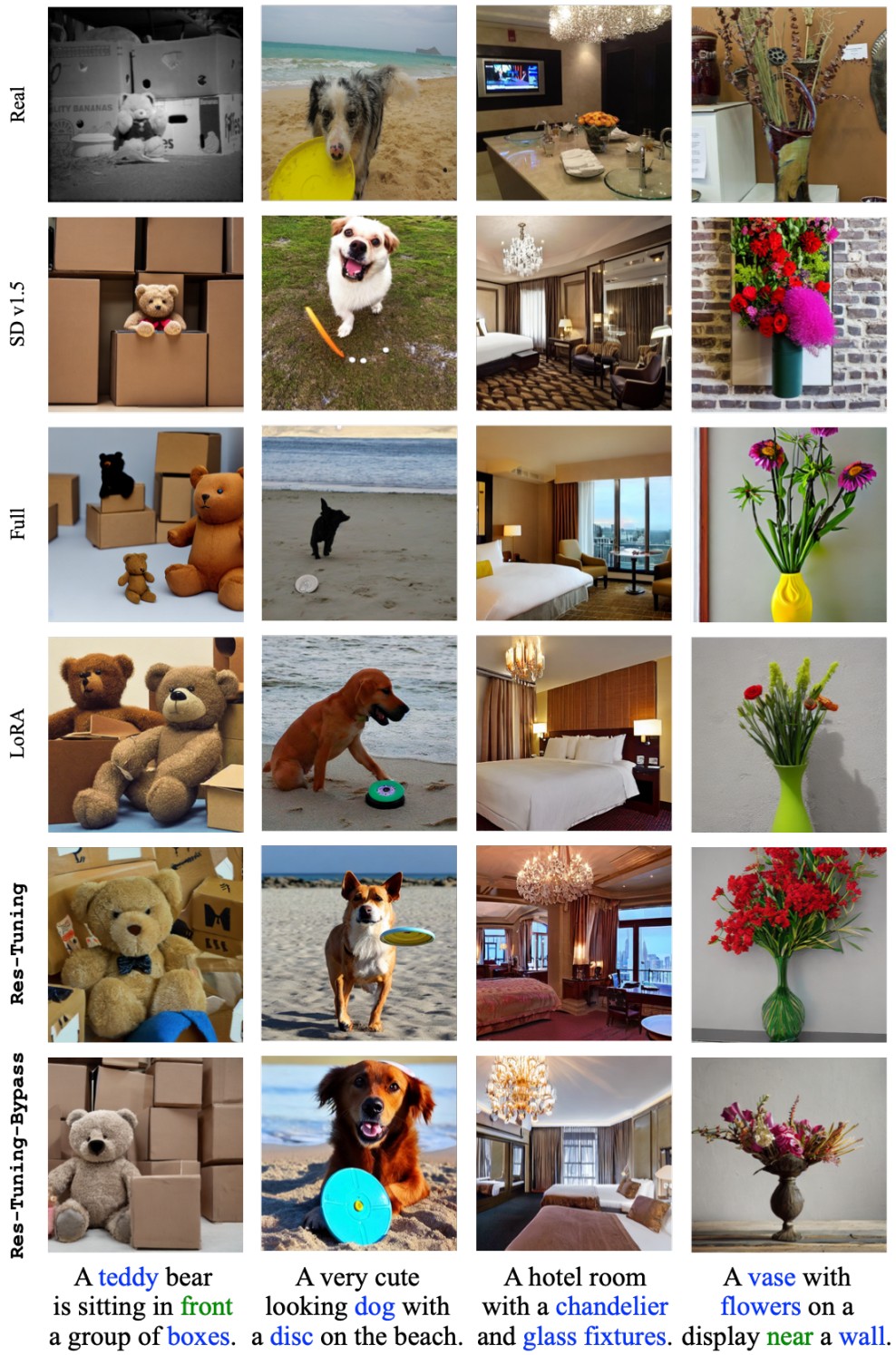

A teddy bear is sitting in front a group of boxes. A very cute looking dog with a disc on the beach. A hotel room with a chandelier and glass fixtures. A vase with flowers on a display near a wall.

Figure 10: Qualitative results of existing tuning strategies and our Res-Tuning on COCO2017 validation set.

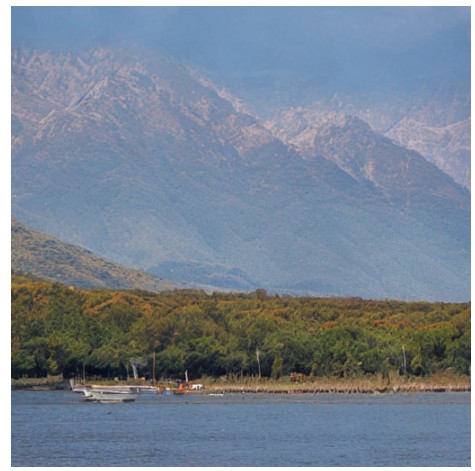

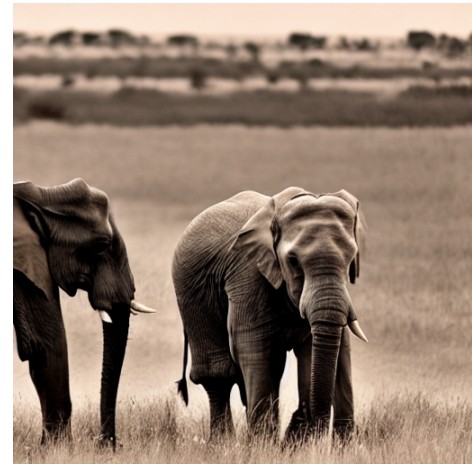

A landscape with water,
a boat, trees and mountains.

Two elephants standing
next to each other in a field.

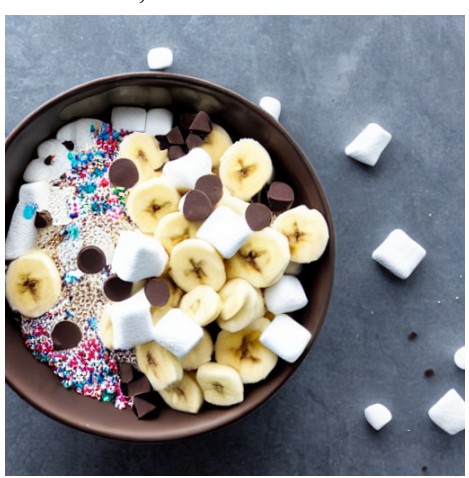

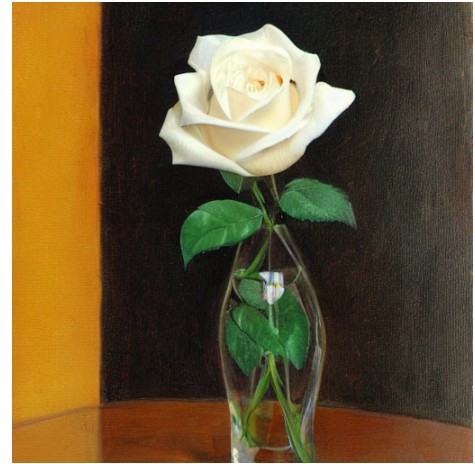

Bananas, marshmellows, chocolate
chips and sprinkles in a bowl.

A single white rose
in a glass vase.

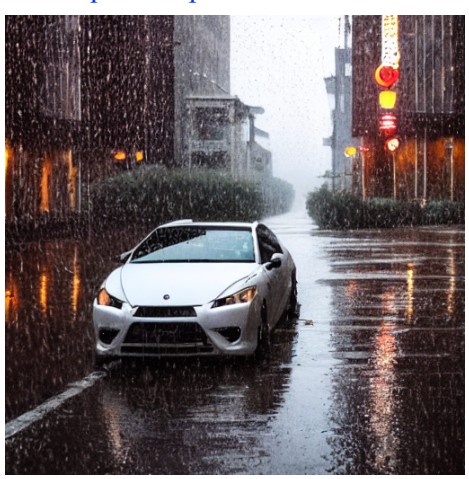

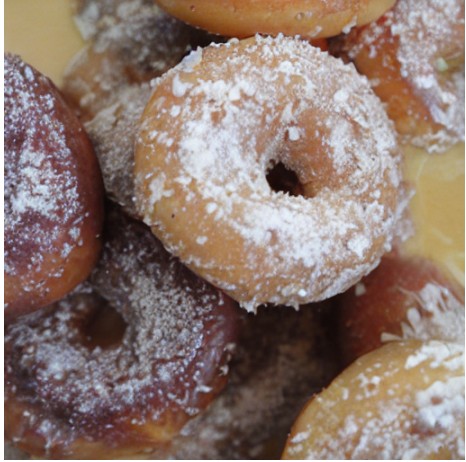

The rain is pouring on
the white car on the street.

A close up of glazed donuts
that are plain or with chocolate.

Figure 11: Visualization of our `Res-Tuning` on COCO2017 validation set.

NABirds                    Stanford Dogs

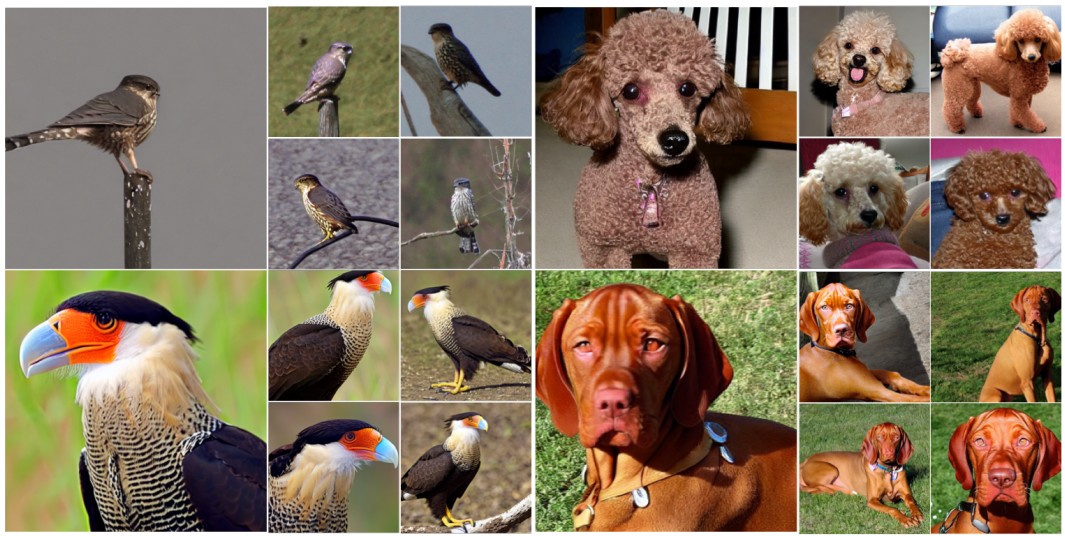

(a) **Res-Tuning**

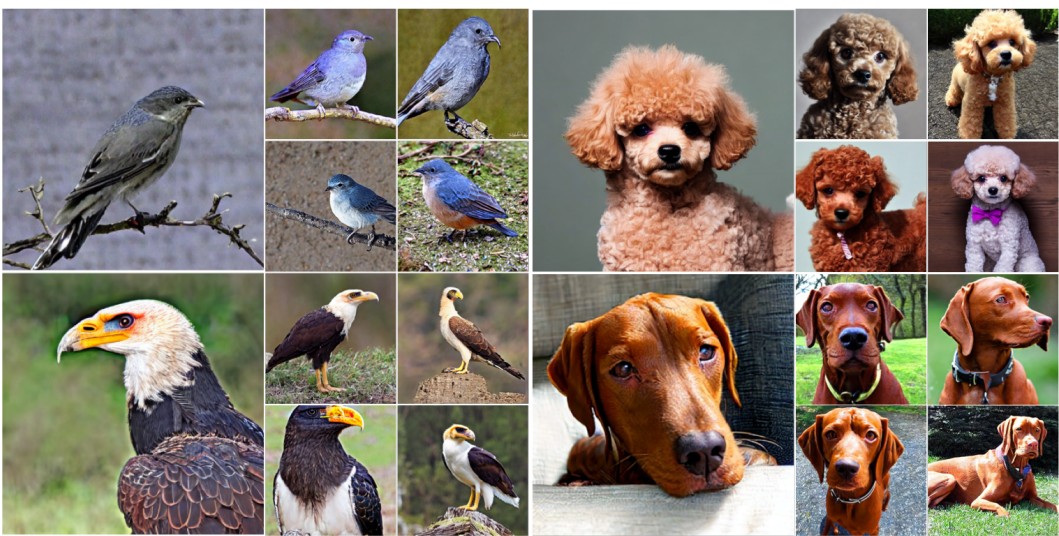

(b) **Res-Tuning-Bypass**

Figure 12: Visualization of our Res-Tuning on NABirds and Stanford Dogs.

Stanford Cars                    Aircraft

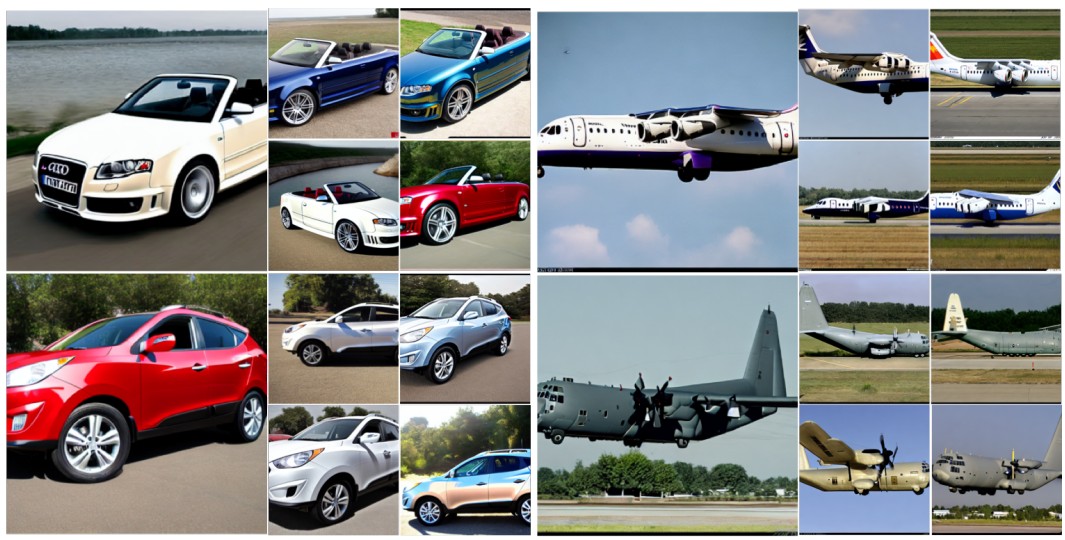

(a) **Res-Tuning**

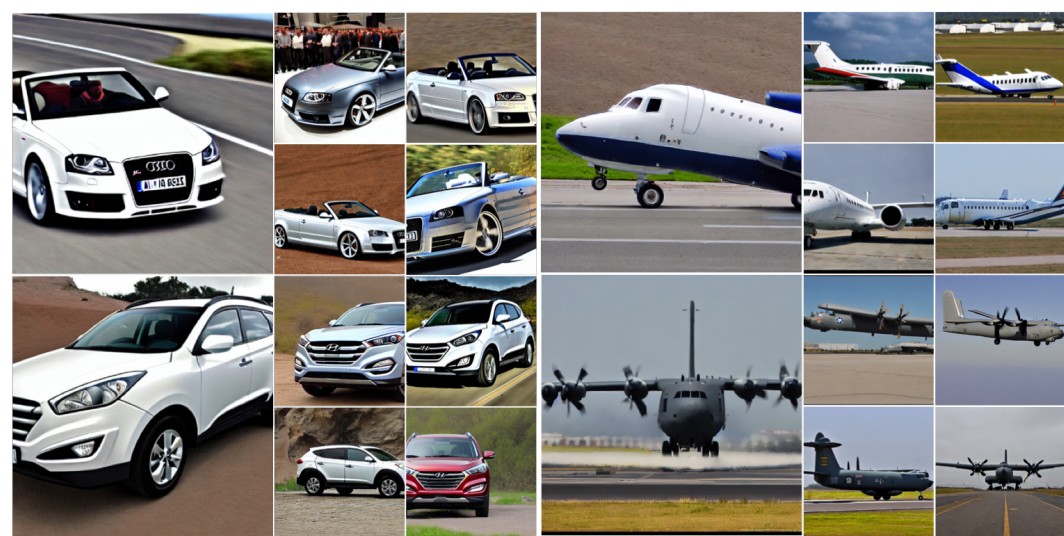

(b) **Res-Tuning-Bypass**

Figure 13: Visualization of `Res-Tuning` on Stanford Cars and Aircraft.

SUN397             Food-101

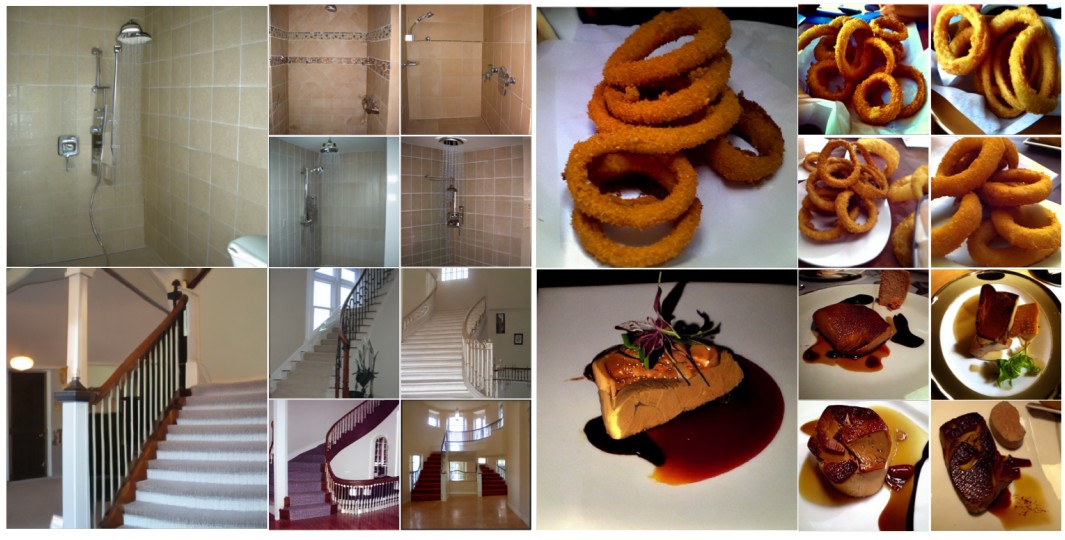

(a) `Res-Tuning`

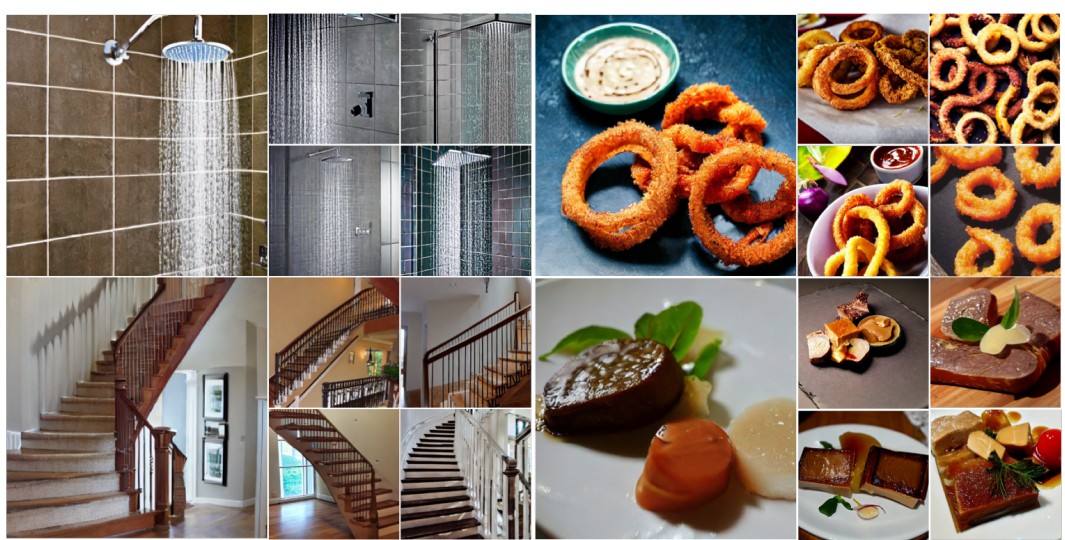

(b) `Res-Tuning-Bypass`

Figure 14: Visualization of `Res-Tuning` on SUN397 and Food-101.

