# OpenReview forum: "Res-Tuning: A Flexible and Efficient Tuning Paradigm via Unbinding Tuner from Backbone"
_NeurIPS.cc/2023/Conference — NeurIPS 2023 poster_

### Official Review · Reviewer_hNoJ · 2023-06-18

**Soundness:** 3 good
**Presentation:** 4 excellent
**Contribution:** 3 good
**Rating:** 6
**Confidence:** 4

**Summary:**

This paper provides a unified framework (called Res-Tuning) to combine different efficient tuning methods. It introduces an unbinding form that integrates existing methods and allows combination flexibility. A memory-efficient variant is also introduced for the sake of training memory efficiency. Experiments are performed on discriminative tasks (CIFAR-100 and VTAB-1K benchmark) and generative tasks (Text-to-image generation on COCO), demonstrating the proposed framework's flexibility and efficiency.



**Strengths:**

1. This paper is well-organized, and the presentation is clear.
2. The formulation of the unified and unbinding form is sound and straightforward and is novel to this field of research.
3. The unbinding form allows combination flexibility, which is the main strength of the proposed methods.
4. Empirical evaluation and theoretical analysis verify the equivalence between the unbinding form and existing approaches.
5. Empirical evaluation on different standard benchmarks shows better performance among existing approaches.

**Weaknesses:**

My main concerns about this paper are the comparisons with previous works.

From my perspective, this paper partially draws inspiration from [13] (MAM-Adapter), which also struggled to unify parameter-efficient tuning methods such as Adapter, Prefix or Prompt tuning. All these methods are evaluated in NLP tasks rather than vision tasks that this paper used.

On vision tasks, VPT [18] is particularly proposed to explore PETL methods in vision (rather than providing a unified view), Thus, the evaluation following VPT is not sufficient for this work. Besides, VPT can work well with Convolution-based foundation models and self-supervised pre-trained models (e.g., MAE, MoCo V3). I don't find such discussion in this work.

**Questions:**

Why does this paper not evaluate the proposed method on NLP tasks that most parameter-efficient tuning methods evaluated on? This paper only includes the reproduced results of MAM-Adapter on CIFAR-100 without any discussion on NLP benchmarks. Since this paper aims to introduce a unified framework for efficient tuning methods, it is necessary to include empirical evaluations on commonly used benchmarks. I would increase my score if sufficient results or discussion were provided.

**Limitations:**

Limitations are not adequately discussed in this paper. It is unknown whether the proposed method can generalize well. Negative societal impacts are unknown.

---

> ### Author Rebuttal · Authors · 2023-08-10
>
>
>
> Dear Reviewer hNoJ,
>
> Thank you for the acknowledgement of our contributions and your valuable comments. We address your concern as follows:
>
> **Q1: Comparisons with previous works**
>
> We would like to first point out that the unified formulation of existing PETL approaches in an unbinding form is only a part of our contribution in the manuscript. The unbinding formulation leads to the Res-Tuning framework, and we discuss its novelty and contribution compared to existing works in detail in the general response.
>
> Another important contribution of our manuscript that distinguishes our work with existing approaches is the memory-efficient variant called Res-Tuning-Bypass, which is able to reduce the resource and time consumption for both training and inference in many scenarios, and is stronger than other existing memory-efficient methods (Side-Tuning and LST) presented in NLP.
>
> **Q2: Extended experiments on different structure**
>
> Following VPT, we provide more analysis on our approach in comparison with the existing PETL approaches. In Table R1, we provide performance comparisons with different pre-training sources. In Table R2, we compare the performances with different convolutional backbones. We will include more relevant results in our revisions.
>
> Table R1. Performance comparison on different pre-trained models on CIFAR-100.
> | Method | MAE | DINO |
> |---|---|---|
> | Full | 85.90 | 87.88 |
> | Linear | 69.83 | 85.51 |
> | Adapter | 85.86 | 89.01 |
> | VPT | 82.44 | 88.33 |
> | Res-Tuning | 86.37 | 89.03 |
>
> Table R2. Performance comparison on different CNN models on CIFAR-100.
> |  | ConvNext |  |  | ResNet-101 |  |  |
> |---|---|---|---|---|---|---|
> | Method | Accuracy | Params(M) | Mem. (GB) | Accuracy | Params(M) | Mem. (GB) |
> | Full | 90.15 | 87.67 | 11.16 | 77.40 | 42.70 | 7.30 |
> | Linear | 90.06 | 0.11 | 3.36 | 54.96 | 0.20 | 2.83 |
> | Res-Tuning | 90.86 | 0.87 | 9.45 | 86.80 | 0.92 | 7.20 |
> | Res-Tuning-Bypass | 90.51 | 1.13 | 3.63 | 72.27 | 3.63 | 4.05 |
>
>
> **Q3: Extended evaluations on NLP benchmarks**
>
> Thanks for the suggestion. Limited by time, here, we preliminarily provide the results of text classification in Table R3. It is observed that the performance of our Res-Tuning framework is on par with or better than MAM-Adapter in text classification, with slightly longer training time but lower memory consumption. Our Res-Tuning-Bypass significantly reduces the training time and memory consumption and achieves a mildly lower performance. In our revisions, we will try to include more NLP-based evaluations for more thorough comparisons.
>
> Table R3. Performance comparison with MAM-Adapter on text classification.
> |  | SST2 |  | MNLI |  |  |  |  |
> |---|---|---|---|---|---|---|---|
> |  | Accuracy | Train (Min/Epoch) | Accuracy | Train (Min/Epoch) | Param. (M) (Train） | Param. (M) (Inference） | Mem. (GB) |
> | MAM Adapter | 94.2 | 7.2 | 87.4 | 41.4 | 46.78 (37.4%) | 0.61 (0.5%) | 22416 |
> | Res-Tuning | 94.56 | 7.9 | 87.45 | 47.3 | 0.97 (0.77%) | 0.97 (0.77%) | 19308 |
> | Res-Tuning-Bypass | 92.94 | 4.2 | 82.01 | 24.2 | 0.98 (0.78%) | 0.98 (0.78%) | 4392 |
>
>
> **Q4: Limitations**
>
> Sorry for the disorientation. We have discussed the limitations and the potential societal impacts in the supplementary material.
>
> [1] Zhang et al. Side-tuning: Network adaptation via additive side networks. ECCV2019.
>
> [2] Sung et al. LST: Ladder Side-Tuning for Parameter and Memory Efficient Transfer Learning. NeurIPS2022.
>
> [3] Jia et al. Visual Prompt Tuning. ECCV2022.

---

> > ### Comment · Reviewer_hNoJ · 2023-08-18
> > **Comment by Reviewer hNoJ**
> >
> > Thank the authors for their feedback. The authors provide sufficient evidence to resolve my concerns. I will raise my score.

---

> > > ### Author Response · Authors · 2023-08-19
> > >
> > > Thank you again for the insightful suggestions that helped improve our manuscript. We really appreciate your adjustment of the rating.

---

### Official Review · Reviewer_DEjH · 2023-06-27

**Soundness:** 4 excellent
**Presentation:** 2 fair
**Contribution:** 3 good
**Rating:** 6
**Confidence:** 4

**Summary:**

This paper proposes a new tuning paradigm, dubbed as Res-Tuning. They first introduce the basic building blocks of foundation models and then unbind three popular tuners from foundation models. They provide theoretical and empirical evidence to support their structural disentanglement. By detaching from the foundation models, they further propose a memory-efficient variant of Res-tuning, dubbed as Res-Tuning-Bypass. They conduct extensive experiments on both discriminative and generative tasks to demonstrate the superiority of their method.

**Strengths:**

- This paper is well-written and the illustration of this paper is concise and easy to understand.
- The idea is simple yet effective.
- They conduct extensive experiments and the results demonstrate superior efficacy and efficiency on both discriminative and generative tasks.

**Weaknesses:**

- There are some detail errors in line 124 and line 130. The reference seems to be Fig.3c, not Fig.3b. And there are some punctuation errors in Eq.(8). There are many similar issues in the article, the author should check this paper again to correct them.
- Code is unavailable now and open-source as soon as possible is beneficial for expanding the influence and credibility of this paper.
- The author provides detailed derivation in supplementary materials, but the manuscript is hard to understand. It would be helpful if the deriving process have a more detailed explanation.

**Questions:**

- Please see the weaknesses part.
- There are no queries in FFN/Block,  How did you apply ResPre/ResPro in FFN/Block?
- The form of adapter tuning in Eq.(6) and Fig. 2c is inconsistent. Is there an FFN before the adapter of the parallel branch?
- LoRa [1] is an effective method in PETL. Adding a comparison with Lora in discriminative tasks can make this paper more solid.
- The full fine-tuning and linear probing results in Tab.3 are inconsistent with the results in SSF[2]. The results of the same task (CIFAR-100) and model (ViT-B/16) are relatively low.

#### Reference

[1] E. Hu, Y. Shen, P. Wallis, Z. Allen-Zhu, Y. Li, L. Wang, and W. Chen. LoRA: Low-rank adaptation of large language models. In Int. Conf. Learn. Represent., 2021.

[2] D. Lian, D. Zhou, J. Feng, and X. Wang. Scaling & shifting your features: A new baseline for efficient model tuning. Adv. Neural Inform. Process. Syst., 2022.

**Limitations:**

The transfer ability of this method depends to a large extent on the performance of the upstream model. This method shares the same vulnerability as existing PETL solutions.

---

> ### Author Rebuttal · Authors · 2023-08-10
>
>
>
> Dear Reviewer DEjH,
>
> Thank you for the acknowledgement of the proposed method and experiments. We address you concerns as follows:
>
> **Q1: Typos.**
>
> Thanks for spotting the errors. We will carefully fix them and polish the writing in our revisions.
>
> **Q2: Code release.**
>
> Limited by our organization's disclosure policy, we are unable to provide the full code for training and evaluating the model for now. But we do have submitted a core part of the model implementation to the AC. Additionally, we are actively preparing the release of the full code and will release them in the near future.
>
> **Q3: Detailed explanation for the deriving process.**
>
> Thanks for the suggestion. We will add more detailed explanation and make corresponding modifications to make the manuscript more easily understandable in our revisions.
>
> **Q4: How is Res-Pre. and Res-Pro. applied in FFN/Block?**
>
> For FFN and Block, we apply them directly with the output of FFN or Block as the query.
>
> **Q5: Inconsistency between Eq.6 and Fig.2c.**
>
> Thanks for pointing this out. In fact, we set out to use the form in Eq.6, but later we opt for the structure in Fig.2c and remove the FFN before the adapter to avoid the backpropagation through the FFN, especially when it is used in the Res-Tuning-Bypass framework. We will clarify this in the revisions.
>
> **Q6: Experiments for the comparison with LoRA.**
>
> Thanks for the suggestion. We will include the following empirical comparisons in the revisions.
>
> Table R1. Performance comparison on FGVC. † denotes our own implementation.
> | Method | CUB_200_2011  | NABirds  | OxfordFlowers  | StanfordCars  | StanfordDogs  | Mean   |
> |---|---|---|---|---|---|---|
> | LoRA† | 86.02 | 80.22 | 99.2 | 85.16 | 88.59 | 87.84 |
> | Res-Tuning | 89.66 | 85.87 | 99.45 | 87.58 | 92.21 | 90.95 |
> | Res-Tuning-Bypass | 88.75 | 83 | 99.61 | 75.41 | 92.4 | 87.83 |
>
> Table R2. Performance comparison on VTAB-1k.
> | Method | Natural Mean | Specialized Mean | Structured Mean | Group Mean |
> |---|---|---|---|---|
> | LoRA | 79.49 | 84.55 | 59.78 | 74.60 |
> | Res-Tuning | 82.29 | 85.46 | 61.19 | 76.32 |
> | Res-Tuning-Bypass | 76.73 | 84.56 | 55.66 | 72.32 |
>
> **Q6: Baseline setting**
>
> For the baseline performance, we mainly followed the settings in AdaptFormer[1] and obtained a similar performance to theirs. We will try reproducing the result of the baseline of SSF later as well as applying the similar experimental settings on our approach to see the result.
>
> [1] Chen et al. AdaptFormer: Adapting vision Transformers for scalable visual recognition. NeurIPS2022.

---

> > ### Comment · Reviewer_DEjH · 2023-08-19
> >
> > Thank you for the author's response. Your answer has addressed my concerns, and I will maintain a positive rating. I am looking forward to a revised version of the paper with fewer errors and the release of the source code.

---

> > > ### Author Response · Authors · 2023-08-19
> > >
> > > Thank you so much for your reply. We sincerely appreciate your recognition and constructive comments to improve our work.

---

### Official Review · Reviewer_nLwc · 2023-07-02

**Soundness:** 3 good
**Presentation:** 2 fair
**Contribution:** 2 fair
**Rating:** 5
**Confidence:** 5

**Summary:**

This paper shows some existing parameter-efficient tuning methods can be decoupled from backbones, which can be formulated as a unified Res-Tuning model. Furthermore, the authors conduct empirical experiments to seek the optimal Res-Tuner. Additionally, a memory-efficient variant of Res-Tuning is introduced by combining outputs of backbone with previous Res-Tuner, which can only compute the gradients for Res-Tuning module. The experiments are conducted on several downstream tasks.

**Strengths:**

+: Compared with parameter-efficient tuning, memory and speed efficient tuning methods show more practical in real-world applications. The proposed Res-Tuning-Bypass show clearly superior to memory-efficient counterparts, i.e., Side-Tuning, LST and linear probing. Meanwhile, Res-Tuning can show speed-efficient on multi-task learning.

+: The proposed method is easy to implement, and is clearly described.


**Weaknesses:**

-: The contribution of this work could be further clarified.

(1)	The statement on unbinding formulation seems a bit overclaimed. For tuning parameters of large models, parameter-efficient modules can be naturally designed in a cascaded or parallel manner. For parallel structures, outputs of frozen backbones and learnable branch can be fused by element-wise addition or multiplication, and such kinds of structures are often used for design network architectures. For Res-Tuning model, it can be regarded as a parallel structure with element-wise addition for fusion. Therefore, it is hardly regarded as a novel or special structure.

(2)	The authors show existing PETL methods have equivalent counterparts in the unbinding formulation, but I feel a bit confused about necessity of this conclusion. Besides, derivation of equivalent on adapter seems be less rigorous. I am not sure could such conclusion help to design Res-Tuner? Because the optimal Res-Tuner seems be decided by empirical experiments.

(3)	I feel a bit confused about the relationship between the unbinding formulation and memory-efficient variant of Res-Tuning. In my opinion, memory-efficient Res-Tuning can be regarded as an improved LST, where an adapter is used for outputs of frozen backbones before fusing with learnable branches. Besides, learnable modules of LST and Res-Tuning are different. The authors would better give more discussion about above issue.

-: Some issues about experiments.

(1)	As shown in Table 3 (c), why Linear probing can be improved when only Bypass is used? The authors would better give more discussion.

(2)	In line 172, improvement of 0.94\% should be corrected to 0.92\%?


-: The writing can be further improved.

(1)	Line 124, Fig. 3b -> Fig. 3c

(2)	FFN$_{adapter}$ in Eq. 6 is inconsistent with description of Fig. 2c.


**Questions:**

Please see paper weaknesses.

---

> ### Author Rebuttal · Authors · 2023-08-10
>
>
> Dear Reviewer nLwc,
>
> Thank you for your time and helpful comments. We address your concerns below:
>
> **Q1: The contribution of the unbinding formulation.**
>
> The essential contribution of the unbinding formulation is to abstract existing PETL methods into a unified formulation of a frozen operation and a learnable tuner. This formulation allows for the **flexible combination** of various approaches, which encompasses existing PETL methods and is able to derive new ones. We would prefer to regard the unbinding formulation as a new perspective towards PETL methods instead of a novel structure.
>
> Additionally, it also serves as the basis for the memory-efficient version Res-Tuning-Bypass, where the side network is entirely constructed by the Res-Tuners in the Res-Tuning framework.
>
> We include more detailed discussions in the general response.
>
> **Q2: Equivalence of unbinding formulation**
>
> The motivation for the unbinding formulation is to provide a unified description to the existing PETL methods. The core reason for us to prove the equivalence between the parallel form and the original form of existing PETL approaches is that the existing PETL methods are proven effective in various tasks, and the equivalence between two forms would ensure the theoretical effectiveness of our formulation. The theoretical proof combined with empirical validation demonstrates that our unbinding formulation is effective and even stronger than existing PETL methods.
>
> **Q3: Relationship between Res-Tuning and Res-Tuning-Bypass.**
>
> Sorry for the disorientation. It is indeed that Res-Tuning-Bypass could be viewed as an improved version of LST. From another perspective, the bypass network in the Res-Tuning-Bypass framework is constructed entirely using Res-Tuners, which is the parallel form of existing PETL methods derived in our unbinding formulation. With both the theoretical proof and empirical validation in Res-Tuning, we can safely reduce the design space and rely on the validated structures for constructing the model. Hence, the Res-Tuning framework and its unbinding formulation serves as an important basis for the Res-Tuning-Bypass model.
>
> **Q4: Issues about experiments.**
>
> (1) Improvement of bypass without Res-Tuners.
>
> Thanks for spotting this. It is indeed an interesting result that worth discussing. We believe the performance improvement brought by the plain bypasses is because the structure of Res-Tuning-Bypass without Res-Tuners essentially performs feature ensemble for features generated by different layers in the Transformer.
>
> (2) Performance improvement of 0.94%.
>
> The number referes to the comparison with existing tuning approaches (underlined numbers in Table 2a), where the highest performance is 92.34%, and our Tri-Res-Tuner, where the performance is 93.28%. Hence, we obtain the least performance improvement of 0.94%.
>
> **Q6: Writing issues.**
>
> Thanks for the suggestion. We will correct the mistakes in our revisions.

---

> > ### Comment · Reviewer_nLwc · 2023-08-18
> >
> > I sincerely thank the authors for providing the feedback. I suggest that the authors could further clarify the relationship between Res-Tuning and Res-Tuning-Bypass and provide the experimental results of plain bypasses without Res-Tuners in the revision.

---

> > > ### Author Response · Authors · 2023-08-18
> > >
> > > Thanks for the valuable suggestion. The relationship between Res-Tuning and Res-Tuning-Bypass will be further clarified in the revision. Furthermore, in our manuscript, we have provided the analysis of our Res-Tuning-Bypass, containing experimental results of plain bypasses without Res-Tuner in Table 3b. We will add more detailed explanation about the results in our revision.

---

### Official Review · Reviewer_Yhf3 · 2023-07-06

**Soundness:** 3 good
**Presentation:** 3 good
**Contribution:** 3 good
**Rating:** 5
**Confidence:** 5

**Summary:**

This paper proposes an unbinding formulation of parameter-efficient methods and further leverage structural disentanglement to develop a memory-efficient variant.
Sufficient experiments on both visual discriminative and text-to-image generative tasks are performed.

**Strengths:**

1. novel implementation.
2. good motivation, which is reasonable for me.
3. good evaluation, promising results, and easy to follow.

**Weaknesses:**

1. The proposed Res-Tuning is not that novel and the analysis of several PET approaches is similar to that in Towards a unified view of parameter-efficient transfer learning (ICLR2022). Parallel adapter design is already a consensus, minor changes seem trivial.
Rather than this boring repetition, I find the memory-efficient variant i.e., Res-tuning-bypass is meaningful and novel. I suggest that the authors make this section a priority and restructure the article.
2. Following Lader Side Tuning, the proposed Res-tuning-bypass can free the backbone from backpropagation. In this way, the training time will be reduced, however, only the training time for generating tasks is reported. Thus I am curious about the training time for visual recognition tasks.
3. The authors compute inference time for multi-tasks, although it seems a bit trivial, I recognize the significance of this time reduction.
However, I cannot accept the comparison with just non memory-efficient methods in Figure 4.  The authors should report LST's results and discuss them carefully.


**Questions:**

The authors employ the CLIP model for text-to-image generation, however, the generative model is more difficult to evaluate its performance.
CLIP-based Parameter efficient approaches have also gained extensive attention recently, such as CoOp[1], PLOT[2], MaPLe[3], and CoPrompt[4]. I am curious whether the proposed Res-tuning-bypass can also reduce the memory cost for vision-language PET approaches with small performance degradation.

[1] Kaiyang Zhou, Jingkang Yang, Chen Change Loy, and Ziwei Liu. Learning to prompt for vision-language models. International Journal of Computer Vision, 130(9):2337–2348, 2022.

[2] Chen G, Yao W, Song X, et al. PLOT: Prompt Learning with Optimal Transport for Vision-Language Models[C]//The Eleventh International Conference on Learning Representations. 2022.

[3] Khattak M U, Rasheed H, Maaz M, et al. Maple: Multi-modal prompt learning[C]//Proceedings of the IEEE/CVF Conference on Computer Vision and Pattern Recognition. 2023: 19113-19122.

[4] Roy S, Etemad A. Consistency-guided Prompt Learning for Vision-Language Models[J]. arXiv preprint arXiv:2306.01195, 2023.


**Limitations:**

Yes, the authors have addressed the limitations.

---

> ### Author Rebuttal · Authors · 2023-08-10
>
>
>
> Dear Reviewer Yhf3,
>
> Thank you for the acknowledgement of the proposed method and experiments. We address your concerns as follows:
>
> **Q1: Novelty of Res-Tuning and similarity to MAM-Adapter in terms of analysis on existing methods.**
>
> We totally agree that the priority of the manuscript is the memory-efficient Res-Tuning-Bypass, and we will reorganize the manuscript to make this clearer.
>
> As mentioned in the general response, the novelty of Res-Tuning lies in its flexibility. It encompasses most popular existing approaches and can derive new ones. Empirically, we found the performance of Res-Tuning stronger than existing PETL methods.
>
> In terms of the similarity to MAM-Adapter with respect to the analysis on existing methods, we claim that our analysis is more rigorous, proving the equivalance of our formulation and existing works both theoretically and empirically. This also provides the foundation for further research on parallel modules for PETL.
>
> Essentially, we believe that Res-Tuning is an indispensible part of our manuscript as it serves as the basis for Res-Tuning-Bypass. Nevertheless, we will adjust the organization of our manuscript to make the emphasis on Res-Tuning-Bypass clearer in our future revisions.
>
> **Q2: Training time on discriminative tasks**
>
> We provide the training time on discriminative tasks in Table R1.
>
> Table R1. The training time on CIFAR-100 corresponds to the Table 2c in manuscript.
>
> | Method | Train (Min/Epoch) | Percentage w.r.t. Full |
> |---|---|---|
> | Full | 2.65 | 100.00% |
> | Linear | 1.19 | 45.10% |
> | MAM-Adapter | 3.07 | 115.72% |
> | AdaptFormer  | 2.18 | 82.23% |
> | Res-Tuning | 2.46 | 92.91% |
> | Side-Tuning | 1.33 | 50.15% |
> | LST | 2.22 | 83.65% |
> | Res-Tuning-Bypass | 1.92 | 72.52% |
>
>
> **Q3: Inference time for multi-tasks**
>
> Thanks for the suggestion.
>
> Figure 4 mainly demonstrates the advantage of Res-Tuning-Bypass framework in terms of inference efficiency on multiple tasks. In fact, such a property is shared by approaches similar to ours, which includes Side-Tuning, LST, as well as linear probing. Including other memory-efficient approaches would make the figure a bit scattered, and it would be almost impossible to distinguish the memory-efficient approaches in the figure. Hence, we will include another figure for comparing the inference efficiency of memory-efficient approaches.
>
> Additionally, we provide a table here for reference in terms of the inference efficiency. Generally, the inference time grows linearly with the number of tasks, so we use the inference time per additional task for comparing the efficiency in Table R2. Overall, the inference time for different approaches is similar. The additional inference time introduced on top of linear probing for Res-Tuning-Bypass is higher than Side-Tuning and lower than LST, but since there are around 10K data for each test dataset, the additional latency introduced by memory-efficient approaches could be negelected (less than 1ms per sample).
>
> Table R2. The testing time on CIFAR-100 between different memory-efficient methods.
> | Method | Testing (sec./test dataset) |
> |---|---|
> | Linear | 23.90 |
> | Side-Tuning | 24.41 |
> | LST | 25.28 |
> | Res-Tuning-Bypass | 24.98 |
>
>
> **Q4: Further evaluations on vision-language task.**
>
>
> We include the results on the vision-language tasks in Figure 2 in the extra pdf material. We noticed that with the same backbone (ViT), the performance is slightly worse than CoOp. In Table R3, we provide the comparison in terms of parameter and the memory consumption between CoOp and Res-Tuning-Bypass. It is observed that the memory consumption of Res-Tuning-Bypass is reduced by 45% when compared to CoOp. In our future revisions, we will include more relevant results and comparisons.
>
> Table R3. Comparison of parameter and memory consumption between CoOp and our Res-Tuning-Bypass.
> |  | Param. (M) | Mem. (GB) | Percentage |
> |---|---|---|---|
> | CoOp | 0.008 | 5768 | 100% |
> | Res-Tuning-Bypass | 0.38 | 3174 | 55% |
>
> [1] Sung et al. LST: Ladder Side-Tuning for Parameter and Memory Efficient Transfer Learning. NeurIPS2022.
>
> [2] Zhou et al. Learning to prompt for vision-language models. IJCV2022.

---

> > ### Author Response · Authors · 2023-08-20
> >
> > Dear reviewer Yhf3,
> >
> > Thanks again for all of your constructive comments and suggestions, which have helped us improve the quality and clarity of this paper!
> >
> > We sincerely hope that our analyses and added experiment on the vision-language tasks could address your concerns. Since the deadline for discussion is approaching, we would like to kindly ask whether there is any additional concerns or questions that we might be able to address.
> >
> > Thanks very much for your effort!
> >
> > Best regards,
> >
> > Authors

---

> > > ### Comment · Reviewer_Yhf3 · 2023-08-21
> > >
> > > I appreciate the rebuttal from the author. The answers have addressed some of my concerns.
> > > So I'm still leaning to accept and keep my initial rating.

---

> > > > ### Author Response · Authors · 2023-08-21
> > > >
> > > > Thank you again for your time and effort in reviewing our manuscript.

---

### Official Review · Reviewer_mYTv · 2023-07-06

**Soundness:** 3 good
**Presentation:** 3 good
**Contribution:** 3 good
**Rating:** 5
**Confidence:** 5

**Summary:**

This paper pays attention to Parameter Efficient Tuning and propose a unified framework namely Res-tuning. More importantly, based on the proposed unified framework, this paper constructs a memory optimization scheme similar to the LST in the language model. Several experiments are performed to validate the effectiveness , including over visual recognition tasks including VTAB-1k, Cifar-100 and text to image generation tasks on COCO and Oxford Flowers and Food-101.


**Strengths:**

1. I am gald to see the success of Res-Tuning-Bypass, an efficient memory PET that is the counterpart of LST in visual tasks.

2. Sufficient experiments are conducted and Res-Tuning achieves impressive performance gain compared to state-of-art approaches.

3. The paper is well structured and easy to follow.

**Weaknesses:**

1. The novelty of unbinded Res-tuning framework seems limited. As for me, Convpass [1] and AdaptFormer adopts a parallel module for PET, which is similar to Res-tuning.

2. The analysis of prompt tuning, prefix tuning, and adapter tuning is similar to that in [2], which is also cited in the paper. For me, the major contribution of this paper is the  Res-tuning-bypass, which brings the success of memory-efficient side tuning to visual tasks.

3. The research of parameter efficient tuning has gained extensive attention recently. More recent related works should be added and described or even compared. In the work, only VPT, SSF, NoAH, and Adaptformer are used as baselines. More recent work such as Convpass[1], SNF[3] should be carefully examined.

4. Lacking some important experiments, such as few-shot experiments on FGVC and Domain Generalization experiments on four ImageNet related datasets (ImageNet-A, ImageNet-R, ImageNet-V2, ImageNet-Sketch).

[1] Jie, Shibo, and Zhi-Hong Deng. "Convolutional bypasses are better vision transformer adapters." arXiv preprint arXiv:2207.07039 (2022).

[2] He J, Zhou C, Ma X, et al. Towards a Unified View of Parameter-Efficient Transfer Learning[C]//International Conference on Learning Representations. 2021.

[3] Wang Y, Shi B, Zhang X, et al. Adapting Shortcut With Normalizing Flow: An Efficient Tuning Framework for Visual Recognition[C]//Proceedings of the IEEE/CVF Conference on Computer Vision and Pattern Recognition. 2023: 15965-15974.

**Questions:**

1. This approach has only been evaluated on ViTs, and I'm curious if Res-tuning-bypass will work on CNNs.

2. The performance of Side Tuning is quite poor, and VPT proves that this memory-saving design is not suitable for visual tasks. This paper replicates the success of LST for the first time in a visual task, and I am very curious about the implementation process. However, the source code is not submitted in the supplementary material. If the authors can provide anonymous github project link in the rebuttal and I am glad to raise my score.

**Limitations:**

Yes, the authors have discussed the limitaions and the potential societal impacts in the supplementary material.

---

> ### Author Rebuttal · Authors · 2023-08-10
>
>
> Dear Reviewer mYTv,
>
> Thank you for the acknowledgement of our contributions and your valuable comments. We address your concern as follows:
>
> **Q1: Novelty of the unbinded Res-Tuning framework.**
>
> As mentioned in the general response, the novelty of the unbinded Res-Tuning framework when compared to parallel modules such as Convpass and AdaptFormer is the flexibility, as well as its ability to derive new PETL methods with the unbinding formulation. Empirically, we also show that Res-Tuning framework performs favourably against them (in Table 3 and Table R1).
>
> More importantly, the significance of the Res-Tuning framework in our manuscript is to provide a basis for the Res-Tuning-Bypass framework. With the unbinding formulation, we can now use the tuners derived from existing PETL methods (proven effective in PETL applications) as the basic building block for the bypass network in Res-Tuning-Bypass.
>
>
> **Q2: Similarity to MAM-Adapter in terms of the analysis of existing PETL methods.**
>
> We agree that the analysis on the existing PETL methods is similar to that of MAM-Adapter. In fact, our Res-Tuning framework is partially inspired by them. However, we believe that our analysis is more rigorous, where we show the equivalence of our framework and the existing methods both theoretically and empirically.
>
> We also agree that the major contribution of the manuscript is Res-Tuning-Bypass. We sincerely thank you for the acknowledgement of our contribution. In the revisions, we will carefully reorganize the manuscript to show that more clearly.
>
> **Q3: More comparison experiments on recent work**
>
> Thanks for the suggestion. Here, we provide an empirical comparison of the mentioned methods in the Table 1 of the extra pdf material. Overall, our Res-Tuning framework shows a competitive performance. We will try to include more recent works in our revision.
>
> **Q4: Extended experiments on Few-Shot Learning and Domain Generalization**
>
> Thanks for the suggestion. We have added the experiments as follows:
>
> - Domain Generalization
>
> Follow the setting of NOAH[4], we first train a model on ImageNet using 16 shots per category and test it on four other variants of ImageNet. The list of data also comes from NOAH and all results are averaged over three random seeds.
>
> Our Res-Tuning goes beyond NOAH by 6.54\% on ImageNet and 2.6\% on the average accuracy of ImageNet. Surprisingly, Res-Tuning-Bypass also achieved better results than other tuning methods.
>
> Table R1. Results on domain generalization. 'Mean' denotes the average accuracy of four variants of ImageNet.
> |  | Source | Target |  |  |  |  |
> |---|---|:---:|:---:|:---:|:---:|:---:|
> |  | ImageNet | IN-V2 | IN-Sketch | IN-A | IN-R | Mean |
> | Adapter | 70.5 | 59.1 | 16.4 | 5.5 | 22.1 | 34.7 |
> | VPT | 70.5 | 58.0 | 18.3 | 4.6 | 23.2 | 34.9 |
> | LoRA | 70.8 | 59.3 | 20.0 | 6.9 | 23.3 | 36.1 |
> | NOAH | 71.5 | 66.1 | 24.8 | 11.9 | 28.5 | 40.6 |
> | Res-Tuning | 78.04 | 66.58 | 29.23 | 13.15 | 29.01 | 43.20 |
> | LST† | 70.00 | 57.04 | 14.39 | 7.21 | 17.02 | 33.13 |
> | Res-Tuning-Bypass | 77.30 | 65.23 | 27.39 | 10.66 | 26.45 | 41.41 |
>
> - Few-Shot Learning
>
> We have included the results for few-shot learning in the extra pdf material. Specifically, we include the few-shot performance on FGVC in Figure 1. Compared with existing parameter-efficient tuning methods, our Res-Tuning shows a certain advantage over the few-shot performance on the FGVC dataset.
>
> **Q5: Extension of Res-Tuning-Bypass to CNNs.**
>
> This is indeed an interesting aspect to explore. We present the results for ConvNeXt pretrained on IN21K and ResNet-101 pretrained on IN1K in Table R2.
> We observe that two convolutional models have different characteristics, where the performance variations of ConvNeXt is small and that of ResNet is large.
> In terms of the effectiveness of Res-Tuning-Bypass, it is observed that it outperforms the fully-finetuned version of ConvNeXt and notably improves the performance of linear probing for ResNet.
>
> Table R2. Results on CNNs backbones.
> |  | ConvNext |  |  | ResNet-101 |  |  |
> |---|---|---|---|---|---|---|
> | Method | Accuracy | Params(M) | Mem. (GB) | Accuracy | Params(M) | Mem. (GB) |
> | Full | 90.15 | 87.67 | 11.16 | 77.40 | 42.70 | 7.30 |
> | Linear | 90.06 | 0.11 | 3.36 | 54.96 | 0.20 | 2.83 |
> | Res-Tuning-Bypass | 90.51 | 1.13 | 3.63 | 72.27 | 3.63 | 4.05 |
>
> **Q6: Code release.**
>
> We are sorry that at the current stage, we are only able to provide the core part of the model implementation as well as relevant documentations (which we have submitted to the AC) due to the disclosure policy on codes within our organization. But we are actively preparing the formal release of the full code for training and evaluating our framework and we promise we will do that in the near future.
>
> [1] Jie et al. Convolutional bypasses are better vision transformer adapters. arXiv.
>
> [2] Chen et al. AdaptFormer: Adapting vision Transformers for scalable visual recognition. NeurIPS2022.
>
> [3] He et al. Towards a unified view of parameter-efficient transfer learning. ICLR2022.
>
> [4] Wang et al. Adapting Shortcut With Normalizing Flow: An Efficient Tuning Framework for Visual Recognition. CVPR2023.
>
> [5] Zhang et al. Neural Prompt Search. arXiv.

---

> > ### Author Response · Authors · 2023-08-20
> >
> > Dear Reviewer mYTv,
> >
> > We would like to thank you again for your time and effort in reviewing our manuscript.
> >
> > It would be greatly appreciated if you could check our responses and provide your valuable feedback. We have given a more detailed explanation about your concerns and provided additional experiments on more SOTA comparisons, few-shot learning on five FGVC datasets, domain generalization on ImageNet and four ImageNet variants, and CNN backbones. This helped us further demonstrate the effectiveness of our work. In addition, we also provide the code implementation for our manuscript.
> >
> > Since the deadline for discussion is approaching, please feel free to let us know if there are any additional clarifications or experiments that we can offer.
> >
> > Best regards,
> >
> > Authors

---

### Author Rebuttal · Authors · 2023-08-10

Dear all,

We would like to express our gratitude to our reviewers for their valuable comments. For positive comments,
- memory-efficiency of Res-Tuning-Bypass (R-mYTv, R-nLwc),
- sufficient and strong experiments (R-mYTv, R-Yhf3, R-DEjH, R-hNoJ),
- well-structured and easy to follow (R-mYTv, R-nLwc, R-Yhf3, R-DEjH, R-hNoJ),
- novel (R-Yhf3, R-hNoJ),
- good motivation (R-Yhf3),
- and flexible (R-hNoJ),

we appreciate them and will carry them forward.

We address the common concern here, which regards the **novelty of the Res-Tuning framework**.

The novelty of the Res-Tuning framework lies in the abstraction of existing PETL methods to the parallel connection of a frozen operation and a learnable Res-Tuner. This allows for the independent development and flexible combination of the structure of the foundation models and the structure of the PETL tuners.

More importantly, the significance of the Res-Tuning framework is that it serves as the basis for the Res-Tuning-Bypass framework. With the unbinding formulation in the Res-Tuning framework, we can treat existing PETL methods as basic building blocks for constructing the bypass network. Such a formulation also allows for the easy adaptation of the Res-Tuning-Bypass model with the development of new PETL methods, which could be simply achieved by replacing the existing Res-Tuners in the bypass network by new modules developed in the future.

*Novelty compared to other existing parallel PETL methods.* Our Res-Tuning framework is able to encompass them and derive new methods based on their module.

*Novelty compared to MAM-Adapter.* The Res-Tuning framework is partially inspired by MAM-Adapter. Besides the modality difference between Res-Tuning and MAM-Adapter, the derivation process for existing PETL methods is more rigorous in Res-Tuning, proving the equivalence of our Tuner with existing methods. We also included the analysis for prompts, which is not included in the analysis of MAM-Adapter.

Indeed, we agree with Reviewer **mYTv**, **Yhf3**, and **nLwc** that the major contribution of our manuscript is the Res-Tuning-Bypass framework, which we will further clarify in the revisions.

For other concerns, we address them in the respective comments to the reviewers.



Thanks and best regards,

Authors of Submission 12594

---

### Comment · Area_Chair_viA5 · 2023-08-17
**Please respond to authors' rebuttal**

Dear Reviewers,

Thanks for your contributions! Please respond to the authors after carefully reading their rebuttals and other reviews. If your assessment of the paper changes, please update your score with a short justification for the new rating.

Thank you,

AC

---

### Decision · Program_Chairs · 2023-09-21

**Decision:**

Accept (poster)

**Comment:**

This paper initially received 5 borderline accepts. The main concerns about this work are: 1) Limited novelty. The comparisons and differences between previous work need to be further highlighted. 2) The comparisons with other SOTA memory-efficient methods (e.g., Ladder Side-Tuning) are missing. 3) The writing quality needs to be further improved. After the rebuttal, all reviewers think most of their concerns have been addressed, and keep a positive rating for this paper. In summary, I recommend accepting this paper, and hope the authors can incorporate all these discussions into the final camera-ready version.